# Long-term drainage reduces $CO_2$ uptake and increases $CO_2$ emission on a Siberian floodplain due to shifts in vegetation community and soil thermal characteristics

M. J. Kwon[1*], M. Heimann[1,2], O. Kolle[1], K. A. Luus[1,3], E. A. G. Schuur[4], N. Zimov[5], S. A. Zimov[5], M. Göckede[1]

[1] Biogeochemical Systems, Max Planck Institute for Biogeochemistry, Jena, Germany

[2] Division of Atmospheric Sciences, Department of Physics, Helsinki University, Helsinki, Finland

[3] Dublin Institute of Technology, Dublin, Ireland

[4] Center for Ecosystem Science and Society, and Department of Biological Sciences, Northern Arizona University, Flagstaff, USA

[5] North-East Science Station, Pacific Institute for Geography, Far-Eastern Branch of Russian Academy of Science, Chersky, Republic of Sakha (Yakutia), Russia

*Correspondence to*: M. J. Kwon (mkwon@bgc-jena.mpg.de)

**Abstract.** With increasing air temperatures and changing precipitation patterns forecast for the Arctic over the coming decades, the thawing of ice-rich permafrost is expected to increasingly alter hydrological conditions by creating mosaics of wetter and drier areas. The objective of this study is to investigate how 10 years of lowered water table depths of wet floodplain ecosystems would affect $CO_2$ fluxes measured using a closed chamber system, focusing on the role of long-term changes in soil thermal characteristics and vegetation community structure. Drainage diminishes the heat capacity and thermal conductivity of organic soil, leading to warmer soil temperatures in shallow layers during the daytime and colder soil temperatures in deeper layers, resulting in a reduction in thaw depths. These soil temperature changes can intensify growing-season heterotrophic respiration by up to 95 %. With decreased autotrophic respiration due to reduced gross primary production under these dry conditions, the differences in ecosystem respiration rates in the present study were 25 %. We also found that a decade-long drainage installation significantly increased shrub abundance, while decreasing *Eriophorum angustifolium* abundance resulted in *Carex* sp. dominance. These two changes had opposing influences on gross

primary production during the growing season: While the increased abundance of shrubs slightly increased gross primary production, the replacement of *Eriophorum a.* by *Carex* significantly decreased it. With the effects of ecosystem respiration and gross primary production combined, drainage increased net $CO_2$ uptake by 25 % in 2013, but decreased it by 35 % in 2014 during the 20 days of the middle of the growing season. These opposing results are due to the very low gross primary production rates in *Carex*-dominated plots in the control areas during 2013, a dry year. In summary, 10 years of drainage increased soil surface temperatures and replaced *Eriophorum a.* by *Carex* sp., which weakened net $CO_2$ uptake rates. During the non-growing season, drainage resulted in four times more $CO_2$ emissions, with high sporadic fluxes; these fluxes were induced by soil temperatures, *Eriophorum a.*, and air pressure.

## 1 Introduction

Arctic ecosystems have long acted as carbon sinks due to their consistent low air temperatures and the presence of permafrost. Although Arctic net primary production and standing biomass are smaller than that of adjacent climate zones (Saugier et al., 2001), the low decomposition rates of Arctic ecosystems due to low temperatures have resulted in an accumulation over 1000 Pg of belowground organic carbon in the upper 3 m of the soil in permafrost regions (Hugelius et al., 2014; Schuur et al., 2015). However, the tendency for Arctic ecosystems to take up more $CO_2$ on average than they release may be changing due to global climate change, which has given rise to shifts in air temperature and precipitation patterns. While the photosynthetic rates and standing biomass in the Arctic have become larger (Epstein et al., 2012; Jia, 2003; Myneni et al., 1997; Xu et al., 2013), the rate of organic carbon decomposition has also increased (Bond-Lamberty and Thomson, 2010), which could potentially accelerate $CO_2$ cycle processes. Further, it is not only a matter of how fast $CO_2$ circulates between the atmosphere and the upper soil layers, but also what will happen to the massive amount of stored

carbon (Schuur et al., 2009). Thus, understanding how $CO_2$ flux patterns of Arctic ecosystems change as a consequence of climate change as well as how this affects the fate of permafrost carbon, is of great importance (Schuur et al., 2008).

Gross primary production (GPP) is chiefly determined by the length of the growing season and leaf area index (LAI); secondary influences are water and nutrient availability, as well as local climate conditions such as air temperature and radiation (Chapin et al., 2012b). As each plant species responds differently to changes in the aforementioned factors controlling GPP, and as successional changes in vegetation species distribution may take place under a changing climate, the total amount of carbon assimilated (*net primary production*, NPP) and plant

respiration (*autotrophic respiration*, $R_a$) may undergo changes. The rate of organic matter decomposition (*heterotrophic respiration*, $R_h$) increases under warmer and more aerobic conditions, and is also influenced by the quality of available organic matter. If any of these conditions are modified due to climate change, the rate of $R_h$ may also change in response.

        Warming air temperatures have been observed in the Arctic (Serreze et al., 2000), and disproportionately

warmer conditions are forecast in response to climate change (Collins et al., 2013; Kirtman et al., 2013; Overland et al., 2014). The influence of temperature on $CO_2$ cycling processes in the Arctic has been well studied (Belshe et al., 2013). As noted in the preceding paragraph, rates of both photosynthesis and organic matter decomposition have increased, and these trends are predicted to continue. This has the potential to change Arctic terrestrial ecosystems from a carbon sink to a source, with accelerated organic carbon decomposition as a dominant process

(Koven et al., 2011; Schaefer et al., 2011). Schuur et al. (2015) predict that, under the current climate warming trajectory, ~5 to 15 % of the permafrost carbon pool may be released into the atmosphere by 2100.

        An increase in air temperature can have an immediate impact on soil hydrology, potentially adding complexity to the drivers of $CO_2$ fluxes and the permafrost carbon pool. In permafrost regions, land surface

warming is usually followed by topographical changes, and thus the formation of small-scale local hydrological conditions: Wetter microsites can form due to subsiding ground following permafrost thaw (Jorgenson et al., 2006; O'Donnell et al., 2011), while adjacent areas become drier as water drains laterally to subsided areas. These phenomena are particularly pronounced when increased air temperature thaws ice-rich permafrost, such as ice wedges and ice lenses (Liljedahl et al., 2016). In some Arctic regions, changing precipitation patterns can aggravate or offset this situation. Precipitation in the Arctic has been generally increasing over the last five decades (Kattsov and Walsh, 2000), but patterns are fluctuating across both time and space (Curtis et al., 1998; Stafford et al., 2000); at times, the surface water balance has also been found to be decreasing (Oechel et al., 2000). Although, overall, greater precipitation is expected in the Arctic as the result of intensified hydrological cycles under climate change, the net effect may significantly vary by region (Bintanja and Selten, 2014; Huntington, 2006; Kirtman et al., 2013). Different areas in the Arctic may therefore become either wetter or drier through the combined effects of atmospheric warming and permafrost thaw, as well as varying rates of precipitation.

Several studies have investigated the effects of drainage on $CO_2$ fluxes in the Arctic (Table 1). Field water table depth (WTD) manipulation experiments and comparison studies with varying WTD have generally shown decreased net $CO_2$ uptake or increased net $CO_2$ emission at lower water levels, primarily due to increased ecosystem respiration (ER; Christensen et al., 2000; Huemmrich et al., 2010; Kim, 2015; McEwing et al., 2015; Oechel et al., 1998; Olivas et al., 2010; Zona et al., 2011); in most cases, GPP increased as well. Although some studies have shown slightly increased net $CO_2$ uptake when the increase in GPP is larger than the increase in ER under drier conditions (Merbold et al., 2009; Natali et al., 2015), the magnitude of the increase in ER is usually larger than that of GPP (Christensen et al., 2000; Huemmrich et al., 2010; Kim, 2015; McEwing et al., 2015; Oechel et al., 1998; Olivas et al., 2010; Zona et al., 2011). The between-site variability of changes in net ecosystem exchange (NEE) presented in Table 1 can be attributed to differences in the observation period,

vegetation type, as well as the intensity and duration of WTD changes in the specific studies. Microcosm experiments have also shown inconsistent results, with a decrease in water level resulting in either decreased (Johnson et al., 1996) or increased (Peterson et al., 1984) net $CO_2$ fluxes. These findings exemplify how the net effect of changes in WTD arise from interactions between multiple factors, and can vary strongly depending on vegetation and soil types (Billings et al., 1982). Therefore, although previous studies have shown that WTD reduction affects GPP and ER rates, the direction and significance of changes in net $CO_2$ cycling have been found to differ from ecosystem to ecosystem.

Most of the existing field observation and incubation studies (Table 1) have focused on the short-term effects of changes in WTD. A further limitation is that most of these studies have been carried out in North America, despite the fact that permafrost regions in Eurasia not only cover about twice the area, but also contain twice the amount of carbon as compared to North America (Tarnocai et al., 2009). Drying manipulation experiments in the Eurasian Arctic with timescales of decades or more will therefore greatly contribute to understanding drainage effects on $CO_2$ fluxes in Arctic ecosystems. In addition to these growing-season $CO_2$ fluxes, several studies have highlighted significant contributions of non-growing season $CO_2$ emissions to the annual $CO_2$ budget in the Arctic (Coyne and Kelley, 1971; Kelley et al., 1968; Panikov and Dedysh, 2000; Webb et al., 2016; Zimov et al., 1993, 1996); however, no studies have yet compared non-growing-season $CO_2$ fluxes between wet and dry sites. Because of the insulation provided by snow, soil temperatures remain warmer compared to air temperatures, and biological processes continue throughout the non-growing season (Kelley et al., 1968; Webb et al., 2016; Zimov et al., 1993, 1996). Non-growing-season fluxes are also affected by state changes from water to ice (Mastepanov et al., 2013).

As a continuation of analysis hydrological manipulation initiated a decade ago in northeastern Siberia (Merbold et al., 2009), the present study investigates how more than 10 years of drainage have affected

ecosystem structure and $CO_2$ fluxes. By directly comparing $CO_2$ fluxes of a pristine area to those from the drained area, our results go beyond a mere description of the immediate disturbance effects, and clearly point out differences between the properties of pristine and drained ecosystems. These differences highlight how the disturbed area has adapted to persistently drier conditions. Our investigation is focused on shifts in soil temperatures, thaw depths (TD), and vegetation community structure, as well as how these changes then influence net $CO_2$ exchange and its component fluxes, GPP and ER (Figure 1). In addition to the growing season, phenomena during the non-growing season will be also described; this represents the first drying manipulation experiment of this nature that extends beyond the growing season.

## 2 Methodology

### 2.1 Site description

The study site is located in a Kolyma River floodplain near Chersky, Northeastern Siberia (also written as Cherskii or Cherskiy). The dominant vegetation species are tussock-forming *Carex* sp. (*appendiculata* and *lugens*) and *Eriophorum angustifolium*. An organic peat layer (15–20 cm deep) has accumulated on top of alluvial material soils (composed of silty clay), although some organic peat materials can be found within alluvial layers due to cryoturbation.

Based on the record filtered by the Berkeley Earth project (berkeleyearth.org, Station ID 169921) for the period 1960–2009, mean monthly temperatures at the Chersky weather station range between -33 ℃ in January and +12 ℃ in July, and the annual mean temperature was -11 ℃. World meteorological organization (WMO) records for the period 1950–1999 indicate a total annual precipitation of 197 mm, with about half of this falling as rain in summer. Snowmelt at the site and in the surrounding river basin usually results in a spring flood. This flooding brings an increased water level of up to 50 cm above the soil surface in late May or early June, followed

by a gradual decrease in the water level starting in early July. After the flood waters have receded, the primary water source is precipitation.

A drainage ring with a ~200 m diameter and minimum depth of 50 cm was constructed in Fall 2004 (Merbold et al., 2009), to drain water into the nearest river channel (Ambolikha). As a result, WTD in this drained area is lowered by 20 cm on average and by up to 30 cm in the growing season compared to control areas (Merbold et al., 2009). While the spatial range of drainage effects varies by soil topography, high-resolution land cover classification (Worldview with 2 m resolution; Richards and Xiuping, 1999) has indicated a high abundance of vegetation groups dominant in dry areas to only within 200 m on both sides of the ditch (Burjack et al., unpublished data); we can therefore limit the drainage effect to this maximum distance. Starting Summer 2013, we measured ecosystem properties and $CO_2$ fluxes at two sites in parallel (Figure 2): the drained area affected by the ditch since 2004 (68° 36' 47" N, 161° 20' 29" E), and a control area (68° 37' 00" N, 161° 20' 59" E) approximately 600 m away from the drained area that is not affected by the drainage ditch. Despite some short-term diurnal fluctuations of up to a few cm following evapotranspiration, as well as precipitation events and water supply from thawing permafrost, distinct differences in WTD between these treatment areas persist over the growing season. Each transect of ten plots in the drained and control areas (henceforth referred to as drained and control transects, respectively) was selected using a stratified systematic sampling method. First, we selected ten approximate positions with 25 m intervals along the boardwalks or transects; we then selected the final plots by considering representative vegetation groups of the selected positions, and by selecting specimens small enough to fit within flux chambers (Table 2, Figure 2). All plots were located within *ca*. 2 m of boardwalks to minimize disturbances.

We conducted three field campaigns. The first was 3 weeks, starting on 20 July 2013 (representing the mid-growing season); the second was 4 weeks, starting on 1 November 2013 (representing the non-growing fall

season); and third was 10 weeks, starting on 15 June 2014 (representing the growing season). The non-growing season was defined as the time period when the average daily air temperature was below 0 ℃. Although WTD of the drained transect was generally lower by 20 cm than that of the control transect after the spring flood in both years, heterogeneous soil topography rendered varying WTD within each transect: One plot in the drained transect had an average WTD close to that of wet plots in the control transect, and two plots in the control transect had an average WTD close to that of dry plots in the drained transect. Since our objective was to analyze how a decade-long drainage disturbance affects $CO_2$ fluxes and its links to environmental parameters, we categorized 20 plots into four groups—drained(D)_high, drained_low, control(C)_high, and control_low— according to transect and WTD category. In 2013, all 20 plots were observed with equal frequency to investigate spatial variability among plots; in 2014, four core plots (i.e., one plot from each group) were more frequently observed to highlight temporal variability over the growing season (Figure 2). Due to different lengths of the observation periods between the two years, we divided data from 2014 into three sub-seasons to distinguish seasonal variability: (2014.1) 15 June–5 July, (2014.2) 6 July–26 July and (2014.3) 27 July–20 August. Sub-season 2014.3 and the 2013 field campaign covered similar periods, based on an analysis of plant phenology with the normalized difference vegetation index (NDVI), and both periods included peak growing season (i.e., when the NDVI of the site was the highest).

## 2.2 $CO_2$ flux measurement

At each plot a 60 cm × 60 cm polyvinyl chloride (PVC) collar was inserted 15 cm into the ground in late June 2013, three weeks before the first flux measurements. No noticeable plant damage was identified around the collars after installation, and three weeks was expected to provide enough buffer time for any stabilization needed in the event of minor belowground damage (Högberg et al., 2001; Parkin and Venterea, 2010). To take the flux measurements, a transparent chamber (60 cm on each side, made of 4 mm-thick plexiglass) was placed on the

collar. The chamber had an opening valve on the top to avoid pressure effects when we placed the chamber onto

the collars. Sensors for air temperature, air humidity, air pressure, and photosynthetically active radiation (PAR) were attached to one side of the chamber and all parameters were measured in parallel with fluxes. These sensors—along with three small fans on a vertical pole attached in one of the corners, for the purpose of mixing the air inside—were placed such that their shadows would not bias incoming solar radiation. $CO_2$ flux was measured with non-steady-state flow-through (i.e., closed dynamic) method using an Ultra-Portable Greenhouse

Gas Analyzer (UGGA, Los Gatos Research, USA), and all data were recorded at 1Hz with a CR1000 data logger (Campbell, USA).

We restricted each flux measurement to two minutes maximum to minimize saturation effects (i.e., warming and pressurized effects) within the chamber. In the event of strong incoming radiation, which can cause temperature to increase more than 1 ℃ per min, we placed ice packs on the collar rims inside the chamber to

keep temperatures constant while measuring fluxes. The number of ice packs was adjusted by observing the temperature changes at 1 Hz frequency. In addition to measuring NEE using the transparent chamber, in summer we also measured ER by covering the chamber with a tarp that blocked incoming radiation. In the non-growing season we did not find significant differences between NEE and ER, probably due to the role of low temperatures, low solar radiation, and snow cover in limiting photosynthesis; we therefore measured NEE only with a

transparent chamber.

To calculate the $CO_2$ flux from the observed changes in $CO_2$ concentrations ($[CO_2]$) within the sampling time of 2 min, median values of the $[CO_2]$ slopes were computed selecting multiple time windows based on a bootstrapping approach, and fluxes (mg C-$CO_2$ m$^{-2}$ s$^{-1}$) were calculated by taking into account air temperature, pressure, and the volume and area of the chamber (Rochette and Hutchinson, 2005). Flux rates that fell outside of

the range of seasonal mean $\pm 3\sigma$ (i.e., standard deviation) were removed as outliers. GPP and ER are expressed in

positive values, indicating the amount of $CO_2$ assimilated and respired, respectively. Negative values for NEE denote net $CO_2$ uptake by the terrestrial ecosystem, while positive values denote net $CO_2$ emission to the atmosphere.

## 2.3 WTD, TD, and soil temperature

WTD was measured during each flux measurement using perforated PVC pipes with a 25 mm diameter, which were installed at each plot. WTD was measured relative to soil surface, with values larger than 0 cm denoting water standing above the soil surface. TD was estimated by pushing a measuring pole into the ground. At every second plot soil temperature probes were installed at 5, 15, 25 and 35 cm (Th3-s, UMS, Germany), and data were recorded while fluxes were measured.

To investigate the effect of soil temperature on $R_h$, we measured respiration rates of soils at 0–15 cm and 15–30 cm depths by aerobically incubating soils at 15 ℃ in the laboratory ($N = 6$ for each depth). Respiration rates were corrected for bulk density and average growing-season soil temperatures at each 0–15 cm and 15–30 cm depth of both the high- and low-WTD plots by assuming a $Q_{10}$ value of 2 as the mean for tundra ecosystems (Zhou et al., 2009). The relative heterotrophic respiration rates between the high- and the low-WTD plots were
subsequently compared, and were linked to changes in soil temperatures.

## 2.4 Vegetation community structure

Changes in vegetation community structure between 2003 (before the drainage ditch was installed) and 2013 (nine years after the drainage ditch was installed) were examined using historical data collected in 2003 through the Terrestrial Carbon Observation System Siberia project (TCOS Siberia; (Corradi et al., 2005).
Vegetation community structure was then identified in 2013 along the same transect as in 2003 (which had not been drained in 2003, but was drained in 2013), as well as in the control transect (newly selected in 2013).

Identification was carried out using the same harvest method in all transects. All living vegetation inside a $1 \times 1$ $m^2$ quadrat ($N = 4$ per transect) was harvested. Collected vegetation was sorted by species, completely dried at 40 °C, and then weighed (g dry biomass $m^{-2}$). Relative abundance of each species (%) was calculated based on the dry biomass to avoid potential biases linked to the water content of plants.

To correlate abundances of plant species with $CO_2$ fluxes without destroying plots for further flux observations, we applied a non-destructive point-intercept method using a $60 \times 60$ $cm^2$ quadrat that was divided into $10 \times 10$ $cm^2$ sub-grids in 2014. After creating this grid, we recorded the plant species that a laser pointer hit when pointed downward at each sub-grid intersection, and calculated the percentage of each species' cover. This analysis was performed within each collar, so that vegetation community structure of each plot could be linked directly to $CO_2$ fluxes. As plots were selected using a stratified method (see Section 2.1), this analysis was also performed at a spot 10 m away from each plot, to confirm that the vegetation community structure of each plot accurately represented the transects.

**2.5 Data analysis and interpolation**

**2.5.1 Interpolation of growing-season $CO_2$ fluxes**

To compare flux variability among plots induced by temporal discrepancies in sampling, and to visualize the implications of these differences for net growing-season $CO_2$ uptake, $CO_2$ fluxes for each vegetation and WTD group were interpolated throughout the growing-season observation period. To simulate $CO_2$ flux rates we adapted a satellite-data-driven $CO_2$ flux model, the Polar Vegetation Photosynthesis and Respiration Model (PolarVPRM), which calculates high-latitude NEE by subtracting GPP from ER (Luus and Lin, 2015):

$$\text{GPP} = \left( \lambda \cdot \text{T}_{\text{scale}} \cdot \text{W}_{\text{scale}} \right) \cdot \text{FAPAR}_{\text{PAV}} \cdot \left( \frac{1}{1 + \frac{\text{PAR}}{\text{PAR}_0}} \right) \cdot \text{PAR} \quad (1)$$

$$T_{scale} = \frac{(T - T_{min})(T - T_{max})}{(T - T_{min})(T - T_{max}) - (T - T_{opt})^2} \quad (2)$$

$$W_{scale} = \frac{a \cdot WTD}{WTD_{max} - WTD_{min}} + b \quad (0 < a < 1, a + b = 1) \quad (3)$$

where $\lambda$ is a parameter representing maximum light use efficiency at low light levels, and $PAR_0$ represents the

half-saturation value of PAR. $T_{scale}$ and $W_{scale}$ are scaling variables ranging between 0 and 1 that reflect the

influence of air temperature and water availability, respectively, on photosynthesis. The set of three parameters

required for calculating $T_{scale}$, i.e. $T_{min}$, $T_{max}$ and $T_{opt}$ were set to 0, 40 and 20 ℃ according to literature

recommendations to avoid the parameter instability that would arise from empirically fitting these parameters,

due to the strong positive correlations between T and PAR. $FAPAR_{PAV}$ is the fraction of PAR absorbed by the

vegetation, and is calculated using the Moderate Resolution Imaging Spectroradiometer (MODIS) Enhanced

Vegetation Index (EVI).

Site-level meteorological observations of air temperature (T) and PAR were used as inputs for

PolarVPRM; these observations were taken from sensors installed in the chamber system (for calibration) and

from nearby meteorological towers (for temporal interpolation; Figure 2). Additionally, ER was calculated as a

$Q_{10}$ function, and the influence of water availability on photosynthesis ($W_{scale}$) was calculated based on WTD

determined next to each plot at the time of flux measurement, with an optimized scaling factor (a, b) to obtain the

best fits between $GPP_{modeled}$ and $GPP_{observed}$.

Both parameters ($\lambda$ and $PAR_0$) were fitted empirically in R (R Core Team, 2013). $PAR_0$ was obtained

from the curve fit between GPP and PAR measured with flux observations using the nonlinear least squares curve

fitting in R (R Core Team, 2013); $\lambda$ was calculated as the slope of the linear regression of observed GPP, and of

GPP calculated from Eq. (1). GPP was estimated excluding PAR terms when no positive relationship between

GPP and PAR was found. GPP was then computed half-hourly using linearly interpolated WTD and EVI, as well as half-hourly measured air temperature and PAR from the meteorological station.

ER was calculated using an empirical $Q_{10}$ model:

$ER = \alpha \cdot e^{k \cdot T}$  (4)

where T is air temperature. The two free parameters in this exponential relationship between ER and air temperature, $\alpha$ and $k$, were empirically calculated from chamber-based measurements of ER and T using nonlinear least squares curve fitting in R (R Core Team, 2013). Once these coefficients were calculated, ER was calculated at half-hourly intervals with half-hourly-averaged air temperature from the meteorological station at

each transect.

Parameter optimization and flux interpolation were carried out separately across four core plots for the year 2014, while 10 plots from each transect were categorized into three vegetation groups and pooled for the year 2013. These vegetation categories took into account only *Carex* sp., *Eriophorum a.*, and shrubs when the relative abundance of each species exceeded 10 % (Table 2). The categorized vegetation groups of the drained

transect were EriophorumShrub, CarexEriophorum and Carex, while those of the control transect were CarexShrub, Eriophorum and EriophorumShrub. The period of interpolation was restricted to the observation periods within each year because WTD ($W_{scale}$) was not measured continuously outside of this period. The discrepancies between the observed and modeled fluxes were calculated using root mean squared error (RMSE) and mean bias error (MBE). All data points that were used for calibration were utilized for the error estimates due

to the limited number of data points.

Uncertainty ranges of parameter fits were calculated using cross validation by creating 2000 data subsets consisting of randomly selected data points (bootstrapping, 80 % of the total dataset). For ER, the 2000 resulting

pairs of parameters were computed for each 1 ℃ temperature bin. A mean ER ± 2σ was estimated to obtain an error range. Similarly, 2000 pairs of $PAR_0$ and $\lambda$ were estimated for binned PAR. The range of GPP was

subsequently estimated by including the rest of the terms from Eq. (1). To constrain the uncertainty ranges of the interpolated fluxes, we took the GPP and ER error ranges at each point from the corresponding PAR and temperature bin, respectively, that reflected the current condition. Because NEE is calculated as the difference of GPP and ER, uncertainty ranges were also determined by adding the two error ranges of GPP and ER. For CarexEriophorum and EriophorumShrub groups in 2013—for which no positive relationship between GPP and

PAR was found or the number of data points was not enough to produce uncertainty ranges, respectively—the bootstrapping step was skipped.

### 2.5.2 Statistical analysis

Spatial differences in the 2013 WTD and TD between the two transects were tested using an independent *t*-test. A permutational multivariate analysis of variance (PERMANOVA) was performed to compare vegetation

community structure between the drained and the control transects of 2013 and 2003. Data from 2014 were not compared with those from 2003 due to the different experimental methods employed. A two-way analysis of covariance (ANCOVA) was carried out with WTD category (high- and low-WTD) and depth as independent variables, to compare soil temperatures between WTD categories. Correlations between WTD and TD were tested by taking values from August of each year when TD was the deepest and the effects of WTD were

strongest.

To see if vegetation groups affected the 2013 fluxes, all fluxes were aggregated by vegetation group (see Section 2.5.1) and a one-way analysis of variance (ANOVA) was performed for each vegetation group as an independent variable. When independent variables significantly influenced dependent variables, Tukey's post hoc test was applied. To investigate whether vegetation group and soil temperatures significantly affected the non-

growing season $CO_2$ fluxes, one-way ANOVA and multiple linear regressions were performed, respectively. A multiple linear regression analysis was also performed to identify additional major environmental drivers for cold-season $CO_2$ fluxes. For multiple linear regression analyses, significant variables were defined based on BIC (Bayesian information criteria); with these selected variables the best-fit regression models were identified, based on the AIC (Akaike information criterion). All statistical analyses were performed using R (R Core Team, 2013).

## 3 Results and discussion

### 3.1 WTD changes from drainage

Following flooding due to snowmelt in early June, the drainage ditch effectively lowered WTD in the drained transect. Average differences in WTD between the two transects were significant with a mean drop of approximately 20 cm, and a maximum difference of up to 30 cm during a three-week period in Summer 2013 (independent $t$-test, $P < 0.001$, $t$ = -4.55, $df$ = 17.91; Figure 3a). Approximately the same difference in mean WTD was observed in the middle of the 2014 growing season. However, several significant rainfall events from late July of 2014 triggered an increase in WTD in the low-WTD plots, especially in the drained transect (Figure 3b). The amount of precipitation was similar at both transects, but WTD for some drained_low plots was more susceptible to increases in WTD compared to the control_low plots; this was because the width of the area within the drainage ring was three times larger than that of the elevated areas of control_low plots. In addition, drainage may slow when the water level rises within the drainage ditch due to the obstruction of water flow by taller vegetation—*Eriophorum a.* and aquatic plants—at the end of the growing season (Allan, 1995; Green, 2005). As a result, WTD in drained_low plots stayed high longer than in the control transect following heavy rainfalls. Similar patterns were also observed in 2005, one year after the drainage ditch was installed (Merbold et al., 2009). Nonetheless, WTD difference between the high- and the low-WTD plots showed distinct patterns. In the long term, it can be speculated that new drainage pathways will be established, which will lead water away more

effectively after precipitation events, and thus reduce the fluctuations in WTD we observed at our site. Transferring our findings to a natural disturbance (e.g., the formation of a connected system of troughs following ice-rich permafrost thaw), we expect that water drainage will be more effective than our drainage manipulation, as thawing permafrost following persistently warmer conditions will induce more pronounced topographical changes (Jorgenson et al., 2006; Liljedahl et al., 2016; O'Donnell et al., 2011).

## 3.2 Shifts in soil temperature and TD and their effects on $CO_2$ fluxes

### 3.2.1 Soil temperature and TD

Our two-way ANCOVA indicated that drainage resulted in both stronger diurnal fluctuations in soil temperatures at shallow layers and colder soil temperatures at deep layers, as compared to the high-WTD plots (Table 3 & Figure 4). This finding highlights the important role of water content in the thermal properties of organic soils, with the soil of shallow layers of the drained transect tending to heat up more easily during the daytime in the low-WTD plots due to the reduced heat capacity of dry organic soil (Abu-Hamdeh, 2003; Idso et al., 1975; Lakshmi et al., 2003; Reginato et al., 1976). At the same time, these dry organic soils also have lower thermal conductivity, limiting downward heat transfer; as a result, deeper layers remained colder than soil at the same depth in the high-WTD plots (Abu-Hamdeh, 2003). This mechanism reduced TD in the low-WTD plots, the effect of which became more distinct at the end of the growing season due to the continued effects of WTD (Figure 5). The positive correlations between WTD and TD in August clearly show this trend (for 2013: $r = 0.47$, $P < 0.05$, for 2014: $r = 0.67$, $P < 0.001$).

### 3.2.2 Soil temperature and TD effects on $CO_2$ fluxes

GPP and ER rates increased with soil surface temperatures (Figure 6) because warmer soil temperatures generally accelerate both photosynthesis (Lawrence and Oechel, 1983; Schwarz et al., 1997) and root respiration

(Boone et al., 1998). The average ER rates of the low-WTD plots were 25 % higher than those of the high-WTD plots in 2013 (independent $t$-test, $P < 0.001$, $t = -5.70$, $df = 532$) despite the fact that GPP rates were found to be lower in the low-WTD plots, meaning that lower $R_a$ rates would have been expected (Figure 6). This increase in the ER rates can be explained by the increased rates of $R_h$ under drier conditions. To make this assessment, we assume that the ratio of total photosynthetic uptake to $R_a$ is constant (Chapin et al., 2012a), although it can slightly vary according to ecosystem type and climate (DeLucia et al., 2007; Zhang et al., 2009), as well as as a result of disturbances (Chapin et al., 2012b).

Modifications in soil surface temperatures had greater impacts on $R_h$ than those in deep soil temperatures. This was to be expected, as decomposition rates are the largest in surface soils, which are more aerobic and contain larger amounts of highly labile organic matter than deeper soils. These factors were further affected by drainage.

Initially, a decrease in WTD following drainage creates more aerobic soil conditions, especially at shallow depths. As anaerobic respiration is slower and less efficient than aerobic respiration, carbon release from both organic and mineral soils (surface and deep soil layers, respectively) under aerobic conditions can be 4–10 times higher than under anaerobic conditions (Lee et al., 2012). Thus, changing from anaerobic to aerobic conditions due to drainage can dramatically increase decomposition rates ($R_h$).

We also found that soils in the drained areas at both depths had similar carbon to nitrogen ratios (C:N), but that there was a greater amount of organic carbon in the surface layers. The C:N is one of the most critical factors determining decomposability (Schädel et al., 2014). Although the litter decay rate is negatively correlated with the C:N in the early stage of degradation (Enriquez et al., 1993), the soil C:N decreases over time as organic matter loses C during decomposition (Malmer and Holm, 1984), and, with lower C:N, more degraded organic matter is impeded from decomposing further (Schädel et al., 2014). In the present study, drainage did not alter

C:N of the surface peat layers, but did increase total carbon content by 11 %, as well as bulk density by 44 % (data not shown). These findings imply that a larger amount of carbon in the surface layers following drainage is susceptible to decomposition. However, when the proportion of shrub litter increases over a longer timescale, reduction in the quantity and quality of litter (Hobbie, 2008) may compensate for elevated decomposition rates ($R_h$).

The warmer soil surface temperatures of the low-WTD plots increased $R_h$ by 240 %, while colder deep soil temperatures reduced $R_h$ only marginally as compared to the high-WTD plots. Combining these two contrasting effects, $R_h$ rates in low-WTD plots were elevated by 95 % as a result of the stronger effects of the soil surface temperatures as compared to deep soil temperatures on compacted peat soils. This increase was largely due to a greater amount of organic carbon—a higher total carbon content as well as more compacted soil—in the

low-WTD plots affected by warmer temperatures at surface layers (as described in the present section). Despite the fact that drainage lowered soil temperatures at deep layers, and, subsequently, TD, the effects of the surface layers were dominant, enhancing the decomposition of organic matter and ER rates. However, the effects of litter quality and quantity, as well as the opposing influence of the physical structures of vegetation on soil temperatures—for example, the negative relationship between shrub abundance and TD due to shade (Blok et al.,

2010) and the positive relationship between shrub abundance and TD due to decreased albedo (Bonfils et al., 2012)—need to be monitored over a longer period of time to gain further insight into the net impact of secondary drainage on carbon accumulation and $CO_2$ fluxes.

### 3.3 Shifts in vegetation community structure and its effects on $CO_2$ fluxes

#### 3.3.1 Vegetation community structure

In its natural, undisturbed state, the vegetation community of this floodplain has historically been dominated by *Eriophorum angustifolium*, followed by *Carex* sp. (*appendiculata* and *lugens*). This vegetation

community structure was reflected in the observations made in 2003 (Corradi et al., 2005)—that is, before the drainage ditch was constructed (Figure 7, top panel)—as well as in the control transect in 2013 (Figure 7, center panel). After a decade of drainage, the abundance of *Eriophorum a.* decreased, while shrubs (*Betula exilis* and *Salix* species, such as *fuscescens* and *pulchra*) and *Carex* sp. became the dominant species in the drained transect (Figure 7, lower panel). While no statistically significant differences were found between the vegetation community structures in 2003 and in the control transect of 2013 (PERMANOVA, $F = 1.62$, $P = 0.19$), significant differences were found between both the 2003 and the drained transect of 2013 (PERMANOVA, $F = 3.31$, $P < 0.05$) and between the two transects of 2013 (PERMANOVA, $F = 5.22$, $P < 0.05$). Although we did not experimentally compare the two observation methods (see Section 2.4), a qualitative comparison of results from 2013 (i.e., harvest) and 2014 (i.e., point intercept) showed a similar abundance of each species; this implies that these two different methods can be used to compare vegetation community structures.

In the control transect the vegetation community structures of the high- and the low-WTD plots were dominated by *Eriophorum a.* and *Carex* sp., respectively, but some low-WTD plots within the drained transect showed a vegetation transition stage. Plots in the drained transect that were categorized as CarexEriophorum showed a mixture of young *Carex* sp. (without discrete tussock forms or small developing tussocks) and short and thin *Eriophorum a.*. The presence of this mixture implies that these areas were formerly dominated by *Eriophorum a.*, which is abundant in saturated areas, but whose abundance decreased due to drainage. In summary, as a result of decade-long drainage, *Eriophorum a.* has largely been replaced by *Carex* sp., and the abundance of shrubs has increased. The core plots that were selected based on WTD category represented this vegetation shift well; control_high and drained_high were dominated by *Eriophorum a.*, control_low was dominated by *Carex* sp. and shrubs and drained_low showed a transition stage from *Eriophorum a.* to *Carex* sp. (Table 2).

We underestimated the abundance of shrubs (*Betula* and *Salix* species) within the collars in the drained

transect as a result of the methodological choice to exclude tall shrubs when selecting plots to ensure that all of

the vegetation could fit into the chambers when measuring fluxes (Note that these results are presented to

compare $CO_2$ fluxes by vegetation group, see Section 2.4). The abundance of shrubs within the collars of the

drained transect was 2 % on average, while independently investigated average abundance along the transect was

20 % on average. This discrepancy will be taken into account in the following sections when interpreting the

420 effects of shrubs on $CO_2$ fluxes.

### 3.3.2 Vegetation effects on $CO_2$ fluxes

Chamber-based $CO_2$ flux measurements during the 2013 growing season showed similar mean and

standard deviations of NEE, GPP, and ER rates between the two transects (Figure 8). However, fluxes showed a

large variability across plots within each transect (each of which was *ca*. 225 m), which results from one-way

ANOVA indicated to be closely linked to the dominant vegetation groups (NEE: $F = 24.99$, $P < 0.001$, GPP: $F =$

11.23, $P < 0.001$, ER: $F = 3.63$, $P < 0.01$; Figure 8). ER also differed by dominant vegetation group, but this

difference was not as pronounced as it was for GPP (Figure 8b & 8c).

The first vegetation effect on $CO_2$ fluxes was that, *Eriophorum*-dominated plots in both transects had

higher rates of photosynthetic uptake than *Carex*-dominated plots. GPP rates of EriophorumShrub were 55 %

higher than those of Carex in the drained transect, and those of EriophorumShrub and Eriophorum were 20 %

higher than those of CarexShrub in the control transect (Figure 8b). Thus, the decrease in *Eriophorum a.* as a

result of drainage reduced carbon accumulation in the terrestrial ecosystem. CarexEriophorum plots in the

drained transect—which represent undergoing a vegetation transition from *Eriophorum a.* to *Carex* sp. following

drainage—showed lower GPP rates than other *Carex*-dominated plots, despite the presence of *Eriophorum a.*

(Figure 8b). In this transition stage (which is here characterized by declining *Eriophorum a.*) or in early

succession stages, plants assimilate less $CO_2$ than they previously did due to lower biomass, and can be more susceptible to disturbances (Chapin et al., 2012c; Niinemets, 2010). The dry year of 2013 was an especially good example of this process: These plots showed slightly decreased GPP rates along with increasing PAR (Supplementary Figure S1), implying that the combination of high PAR and high air temperature caused water stress to plants. Taking into account such transition effects after 10 years of drainage is important given that the fraction of these areas of the total area—three out of ten plots—is not small. Moreover, this finding highlights the fact that ecosystem adaptation to new environmental conditions may take a long time, as 10 years was evidently not sufficient for plants to stabilize in this new, drier condition. Carex and CarexShrub—which can be considered the potential vegetation communities of drained_low plots—showed slightly higher GPP rates than CarexEriophorum in the drained transect, and no decreasing GPP rates with higher PAR in both 2013 and 2014. However, CarexShrub in the control transect took up significantly less $CO_2$ under the drier conditions of 2013 than in 2014; this remained true despite the fact that GPP rates increased along with PAR, suggesting these plots were also experiencing dry conditions. This implies that when *Eriophorum a.* is fully replaced by more drought-tolerant species and vegetation community structures are stabilized to a persistently dry condition for a longer time period, CarexEriophorum plots will take up more $CO_2$, and may not undergo water stress as easily as they currently do. Still, overall, Carex and CarexShrub showed lower GPP than high-WTD *Eriophorum*-dominated plots, indicating that drainage can potentially reducing carbon assimilation through the replacement of *Eriophorum a.* by *Carex* sp..

Increasing shrub abundance can compensate for lowered GPP rates in drained areas following a reduction in *Eriophorum a.* coverage. EriophorumShrub of the control transect, which had 10 % shrub coverage, had, on average, 4 % higher GPP rates than Eriophorum in 2013 (although this difference was not significant; Figure 8b). This difference is expected to be larger following drainage, as average abundance of shrubs in the drained transect was 20 %—twice as large as in the control transect. If GPP rates increase proportionate to the abundance

of shrubs, GPP rates in the drained transect may be approximately 8 % larger than measured. While this compensation effect by shrubs is less than the effect of replacement of *Eriophorum a.* by *Carex* sp. in 2013, this effect can be larger in less dry conditions, as CarexShrub took up slightly more $CO_2$ than EriophorumShrub in the control transect during the warmer and wetter year of 2014 (Table 2). These differences suggest that increasing the abundance of shrubs can lead to higher $CO_2$ assimilation rates, depending on climate conditions.

Other factors can also influence the productivity of shrubs, such as nutrient availability and active layer depths in Arctic ecosystems (Chapin et al., 2012a; Myers-Smith et al., 2011). Shrubs generally assimilate a larger amount of carbon than sedges (Shaver and Jonasson, 2001), and GPP may increase in the drained area, as a result of both larger and more numerous shrubs. Moreover, increasing the abundance of shrubs not only changes carbon exchange rates between the atmosphere and the terrestrial ecosystem, but also carbon storage patterns within the terrestrial ecosystem. In the drained transect, living aboveground biomass—the sum of leaf and stem—was larger than in the control transect, and in 2003, while the biomass of green leaves decreased, that of stems increased, mostly due to the increased abundance of shrub species (Supplementary Figure S2). When shrubs continue to expand, a large portion of carbon will be stored in plants, especially in shrubs' stems, and the amount of litter added to the soil will decrease accordingly (Supplementary Figure S2).

In the present study, after a decade of drainage, the effects of decreasing *Eriophorum a.* were generally larger than those of increasing shrub abundance. This could at least in part reflect a transitional vegetation stage characterized by reduced biomass and underestimated shrub abundance. However, the effects of shrubs can become larger as their abundance increases, or can temporarily increase when increased precipitation counterbalances dry conditions. In addition, wet floodplain ecosystems with large annual peat accumulation in the soil will develop into dry ecosystems with more carbon stored in aboveground standing biomass upon drainage (Shaver and Jonasson, 2001).

### 3.4 Growing-season $CO_2$ fluxes

### 3.4.1 Gap-filled growing-season $CO_2$ fluxes

The modeled fluxes for both 2013 and 2014 had similar patterns to the observed fluxes: *Eriophorum*-dominated plots (i.e., high-WTD plots) generally showed higher GPP rates than *Carex*-dominated plots (i.e., low-WTD plots) in both transects (Table 4). In addition, the 10 % difference in shrub cover between Eriophorum and EriophorumShrub from the control transect did not significantly affect GPP rates (Table 4). In 2014, the rates of GPP were low at the beginning of the growing season, increasing by mid-growing season as air temperature, PAR, and plant biomass increased. Variations among groups in 2014 were similar to those in 2013, except for slightly higher GPP rates in *Carex*-dominated plots than in *Eriophorum*-dominated plots in the control transect. The increased rates of GPP in *Carex*-dominated plots in 2014 can be attributed to wetter conditions following increased precipitation. ER rates were consistently greater in the low-WTD plots, in part due to increased $R_h$ rates, and the cumulative ER rates increased with drainage by 5 % in 2013 and by 10 % in 2014 (Table 4).

Combining the effects of vegetation and soil temperatures on GPP and ER rates, the net effects of drainage on $CO_2$ fluxes (NEE) was -0.3 g C-$CO_2$ m$^{-2}$ day$^{-1}$ (i.e., 25 % more $CO_2$ uptake) in 2013 and +0.98 g C-$CO_2$ m$^{-2}$ day$^{-1}$ (i.e., 35 % less $CO_2$ uptake) in 2014 when daily $CO_2$ fluxes of 20 days, weighted by the number of plots of each group, were compared (Table 4). This range was comparable to other drainage studies presented in Table 1. However, the net $CO_2$ flux changes were in opposite directions in these two years due to the control_low plots' sensitivity to climate conditions during the 2013 and 2014 observation periods. In 2014, air temperatures were 8 ℃ warmer and soils were wetter than in 2013, leading to a smaller difference in WTD between the two transects (14 cm in 2013 and 22 cm in 2014). As described in Section 3.3.2, more precipitation in 2014 may have offset dry conditions in control_low plots; however, drained_low plots may not have been affected by this increase in precipitation due to the transitional low-biomass vegetation stage. Despite the variability between

years for transect level, there were consistent trends in both years: After 10 years of drying manipulation, the replacement of *Eriophorum a.* by *Carex* sp., more aerobic conditions, and increased soil surface temperature all weakened net $CO_2$ uptake (Table 4).

### 3.4.2 Model error from interpolation

Comparing observed against modeled flux rates for all individual measurements in the database, the mean RMSE of ER was 0.009 and 0.007 mg C-$CO_2$ m$^{-2}$ s$^{-1}$ for 2013 and 2014, respectively, and that of GPP was 0.021 and 0.016 mg C-$CO_2$ m$^{-2}$ s$^{-1}$ for 2013 and 2014, respectively (Table 5). The control transect in 2013 had generally lower RMSE and MBE compared to the drained transect (Table 5). Larger RMSE and MBE in the drained transect can be attributed to the pooling of data points by vegetation group, as well as to the limited number of data points; the large error in GPP for the Carex group of the drained transect can be attributed to varying standing biomass, and that of the EriophorumShrub in the drained transect stems from the small number of data points. In 2014, data points for each group came from only a single plot, but varying WTD and thickening thaw depth over the growing season resulted in relatively large errors (Table 5).

The PAR terms were excluded when fluxes of CarexEriophorum from the drained transect were interpolated. This choice was made because plotting GPP against PAR sorted by vegetation group indicated that in 2013 GPP rates increased with increasing PAR in all but CarexEriophorum of the drained transect (Supplementary Figure S1). This plot showed slightly decreasing GPP with increasing PAR in the relatively dry year of 2013, as described in Section 3.3.2., implying that high PAR and air temperatures in combination with dry conditions may have induced water stress. This water stress then had a net effect of suppressing rather than enabling photosynthesis under warm and sunny conditions.

Low uncertainty ranges imply that variations in ER and GPP can be mainly explained by air temperature and PAR, respectively. The uncertainty ranges of ER were large compared to those of GPP (Table 5), suggesting that ER rates varied with factors other than air temperature, while GPP rates mostly varied with PAR.

### 3.5 Non-growing-season $CO_2$ fluxes

Due to the low air temperatures and weak solar radiation, GPP in the non-growing season was negligible. Although a limited amount of photosynthetic activity could have theoretically taken place during this time (Atanasiu, 1971), no significant differences were found between NEE and ER, implying that $CO_2$ fluxes consisted mostly of $CO_2$ released from the soil. The drained transect emitted an average of four times more $CO_2$ than the control transect (Figure 9). If the observed flux pattern is representative for the entire month of November, the net $CO_2$ emission of this month would be 11 g C-$CO_2$ m$^{-2}$ in the drained transect and 3 g C-$CO_2$ m$^{-2}$ in the control transect.

Some plots in the drained transect showed sporadically high fluxes, the rates of which were comparable to ER rates from the growing season (Figure 9). These high fluxes in the drained transect could be linked to vegetation groups, especially the abundance of *Eriophorum a.*, as well as to air pressure and soil temperatures (multiple linear regression, adj. $R^2 = 0.46$, $P < 0.001$; pressure, $P < 0.001$; Tsoil at 5 cm, $P < 0.001$; *Eriophorum*, $P < 0.001$; pressure × Tsoil at 5 cm, $P < 0.001$; pressure × *Eriophorum*, $P < 0.001$; Figure 9). This may be a part of the physical processes outlined by Mastepanov et al. (2008, 2013), through which the freezing of soil pushes stored $CO_2$ and $CH_4$ gases in soil to the atmosphere through cracks in soil or dead plant bodies. Although soil temperatures between 0–35 cm were consistently below zero, soil temperatures at 35 cm did not fall below -5 ℃ until the end of November. Ongoing freezing at greater depths than 35 cm and low air pressure could have stimulated $CO_2$ emission from the soil to the atmosphere through dead *Eriophorum a.*. The fact that the $CO_2$ fluxes were influenced by soil temperatures implies that high $CO_2$ emissions were not exclusively triggered by

the physical expression of existing $CO_2$ in soils, but also from ongoing respiration at relatively mild soil temperatures insulated by snow (Kelley et al., 1968; Webb et al., 2016; Zimov et al., 1993, 1996). $CO_2$ fluxes in the control transect were also influenced by the abundance of *Eriophorum a.*, air pressure, and soil temperatures (multiple linear regression, adj. $R^2 = 0.21$, $P < 0.001$; Tsoil at 5 cm, $P < 0.01$; Tsoil at 15 cm, $P < 0.01$; Tsoil at 25 cm, $P < 0.01$; *Eriophorum*, $P < 0.05$; Tsoil at 5 cm × *Eriophorum*, $P < 0.001$; Tsoil at 15 cm × *Eriophorum*, $P < 0.001$; Tsoil at 25 cm × *Eriophorum*, $P < 0.001$), but the rates were relatively constant over time and without high sporadic fluxes, unlike in the drained transect.

Although the $CO_2$ fluxes in the non-growing season were partially explained by vegetation group, air pressure, and soil temperatures, the amount of variation ($R^2$) together explained by these factors was low. We also cannot firmly conclude that the observed sporadic high $CO_2$ fluxes in November were largely driven by these factors, because we did not observe $CO_2$ fluxes continuously along with soil temperatures; what's more, these high $CO_2$ fluxes were only observed in the drained transect despite there being similar conditions in the control transect. High uncertainties and limitations in predicting both non-growing-season $CO_2$ fluxes and possible high $CO_2$ fluxes during the thawing season (Friborg et al., 1997) need to be addressed to determine the net effects of drainage on the annual $CO_2$ fluxes of this site. Nevertheless, the observed considerably higher $CO_2$ fluxes in the non-growing season for the drained transect imply that drainage not only affects growing-season $CO_2$ fluxes, but also has the potential to alter non-growing-season fluxes significantly.

**5 Conclusion and final remarks**

Drainage of a floodplain near Chersky resulted in an average WTD drop of 20 cm. This, substantially altered both biogeophysical and biogeochemical ecosystem properties over the span of a decade, with profound net impacts on $CO_2$ fluxes. The first change important for $CO_2$ processes was that vegetation community structure in drained areas shifted significantly toward increased *Carex* sp. and shrub species (*Betula* and *Salix* species) and

decreased *Eriophorum a.*. The second change was that WTD variation led to divergent soil temperature profiles by depth, with drained areas showing greater fluctuations in soil surface temperatures due to their low heat capacity, and with deeper soil demonstrating colder temperatures due to the low thermal conductivity of the dry

soil above it. Consequently, the drained areas had shallower thaw depths compared to the control areas.

These aboveground and belowground changes significantly affected $CO_2$ fluxes. The drained areas showed higher ER due to more aerobic conditions, with a greater amount of organic carbon affected by warmer soil surface temperatures. Dominant plant species in the drained areas took up less $CO_2$ (i.e., *Carex* sp. engaged in less GPP) than *Eriophorum a.*, which is dominant in the control wet areas. Increased abundance of shrubs

slightly compensated for the decrease in GPP, but, in our datasets, it could not fully balance out the losses. Overall, drainage increased net $CO_2$ uptake (NEE) by 25 % in 2013 but decreased it by 35 % in 2014 during the 20 days of the growing season when the two transects were compared. The opposite patterns of the two years can be attributed to the control_low plots, which showed large variations with climate. Despite the inter-annual variability, both years had consistent trends toward the replacement of *Eriophorum a*. with *Carex* sp., more

aerobic conditions, and increased soil surface temperature, all of which weakened net $CO_2$ uptake (NEE). In the non-growing season, $CO_2$ emission was four times larger in the drained than in the control areas, partially as a result of the abundance of *Eriophorum a.*, air pressure, and soil temperatures.

Ecosystem changes after 10 years of drainage in an Arctic floodplain decreased $CO_2$ uptake and increased $CO_2$ emissions in both the growing and non-growing seasons. These findings highlight the importance

of considering the changes in ecosystem properties under persistent dry conditions when investigating $CO_2$ fluxes in response to global climate changes. As ongoing global warming thaws ice-rich permafrost and makes some regions drier, Arctic wetlands may accumulate less carbon in the terrestrial ecosystem, respire more $CO_2$ from shallow soil layers, and preserve carbon in deep soil layers. Given that vegetation communities continue

changing after 10 years, with different areas then responding differently to climates, further observations of this site, as well as of other ecosystems in the Arctic, are needed over a longer term to better predict the fate of the Arctic in the face of global climate changes.

**Acknowledgements**

This work has been supported by the European Commission (PAGE21 project, FP7-ENV-2011, Grant Agreement No. 282700, and PerCCOM project, FP7-PEOPLE-2012-CIG, Grant Agreement No. PCIG12-GA-2012-333796), the German Ministry of Education and Research (CarboPerm-Project, BMBF Grant No. 03G0836G), the International Max Planck Research School for Global Biogeochemical Cycles (IMPRS-gBGC), and the AXA Research Fund (PDOC_2012_W2 campaign, ARF fellowship M.Goeckede). The authors wish to express their appreciatiation to NESS staff members, especially Galina Zimova, Nastya Zimova and Vladimir Tatayev for organizing and assisting with field work; Chiara Corradi and Lutz Merbold for giving us valuable advice; Martin Hertel, Frank Voigt, Waldemar Ziegler, and other Freiland group members for technical support; Ina Burjack for providing an aerial map and growing-season partitioning scheme with vegetation phenology, as well as for assisting with field and lab work; Marcus Wildner, Carsten Schaller, and Fanny Kittler for assisting with field work; Mirco Migliavacca for advising on data analysis; and Ines Hilke and other RoMA group members for soil analysis. We ordered our authors according to both the first-last-author-emphasis and equal-contribution (i.e., alphabetical sequence) methods (Tscharntke et al., 2007).

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

Table 1 CO$_2$ flux changes (g C-CO$_2$ m$^{-2}$ day$^{-1}$) in response to a water table depth (WTD) decrease, expressed as

either flux$_{control}$ − flux$_{lower-WTD}$ or flux$_{higher-WTD}$ − flux$_{lower-WTD}$. Negative net CO$_2$ flux rates represent a net increase

in terrestrial CO$_2$ uptake; positive changes denote a decrease in net CO$_2$ uptake by the terrestrial ecosystem or an

increase in terrestrial CO$_2$ emissions to the atmosphere. The ranges of these changes are from different years, soil

types, and study sites. Numbers in parentheses represent percent change compared to the original (control, WTD

condition) flux.

| Sites | WTD change | Net CO$_2$ flux change | Reference |
|---|---|---|---|
| Coastal plain | Drawdown | + 0.59 (+160 %) | Huemmrich et al. (2010) |
| | 3 cm lower | + 0.23 (+87 %) | Olivas et al. (2010) [1] |
| | Up to 3.6 cm lower [6] | + 1.17 (+63 %) | Christensen et al. (2000) [2] |
| | 7–7.5cm lower | + 0.36 to 0.4 (+450 % to 500 %) | Oechel et al. (1998) |
| | 8.5 cm lower | + 2.99 (+365 %) | Kim (2015) [3] |
| | 11.9 cm lower [6] | + 0.41 (+67 %) | Zona et al. (2011) [3] |
| | 20 cm lower [6] | + 0.72 (+37 %) | McEwing et al. (2015) |
| Floodplain | 20–35 cm lower | - 0.06 (-47 %) | Merbold et al. (2009) [4] |
| Moist tundra | 2.5 cm lower | - 0.02 (-3 %) | Natali et al. (2015) [5] |
| Laboratory | Saturated | - 2.63 to - 1.41 | Johnson et al. (1996) |

| | | |
|---|---|---|
| vs. field capacity | (-1716 to -344 %) | |
| 5 cm lower | - 0.61 to + 0.96 | Billings et al. (1982) |
| | (-59 to +72 %) | |
| 10 cm lower | + 2.21 (+184 %) | Peterson et al. (1984) |

[1] Only data from 2008 were used, as this was the only time when the WTD of the drained area was lower than that of the control area.

[2] Only from grassland data.

[3] Only ecosystem respiration was considered (no gross primary production).

[4] Only data from 2003 and 2005 were used, as these were the only years when climate conditions were similar.

[5] Only data from 2013 were used, as this was the only time when the WTD of the drained area was lower than that of the control area.

[6] WTD difference from natural variation instead of manipulation.


Table 2 Vegetation and water table depth (WTD) characteristics of plots. Vegetation groups were created when the relative abundance of each species exceeded 10 %. Average WTD was calculated by pooling all WTD measurements from both years by each vegetation group (mean ± standard deviation), except the period where the whole area was flooded from snowmelt. When the average WTD of the growing season was larger than -10 cm, plots were classified as high-WTD group.


| Transect | Plot ID Number | Vegetation group | Average WTD (cm) | WTD group |
|---|---|---|---|---|
| Drained | 0 | EriophorumShrub | 4.6 ± 2.2 | High |
| | 1, 2, 4 | CarexEriophorum | -14.1 ± 8.4 | Low |
| | 3, 5, 6, 7, 8, 9 | Carex | -19.2 ± 6.1 | Low |
| Control | 0 | CarexShrub | -1.3 ± 2.3 | High |
| | 1, 3, 6, 7, 8, 9 | Eriophorum | 4.3 ± 2.4 | High |
| | 2 | EriophorumShrub | 3.9 ± 2.1 | High |
| | 4, 5 | CarexShrub | -18.5 ± 4.1 | Low |

Table 3 Two-way ANCOVA results with water table depth (WTD) category (high and low), and soil depth (cm) as the independent variables and soil temperature (℃) as the dependent variable. The time periods of the entire year of 2013, as well as three sub-seasons of 2014—(2014.1) 15 June–5 July, (2014.2) 6 July–26 July and (2014.3) 27 July–20 August—were separately analyzed. The significance of $F$ values are denoted with asterisks ($P$ value $< 0.001$ ***, $< 0.01$ **, $< 0.05$ *).

|  | Transect | WTD | Depth | WTD×Depth |
|---|---|---|---|---|
| 2013 | Drained | 12.75 *** | 602.64 *** | 13.38 *** |
|  | Control | 15.38 *** | 700.93 *** | 2.64 |
| 2014.1 | Drained | 3.54 | 169.46 *** | 0.02 |
|  | Control | 32.55 *** | 165.35 *** | 29.56 *** |
| 2014.2 | Drained | 26.21 *** | 400.48 *** | 0.52 |
|  | Control | 1.24 | 380.91 *** | 2.42 |
| 2014.3 | Drained | 101.87 *** | 680.50 *** | 7.55 ** |
|  | Control | 6.49 * | 813.62 *** | 4.91 * |


Table 4 Average daily flux (g C-CO$_2$ m$^{-2}$ day$^{-1}$) from interpolation for the period of 22 July to 10 August (20 days) in both 2013 and 2014. Values in parentheses are cumulative flux (g C-CO$_2$ m$^{-2}$) for the period of 22 July to 10 August (20 days) in 2013 and 16 June to 20 August (66 days) in 2014. Results from 2013 represent the fits of all data points ± standard deviation from bootstrapping, and those of 2014 represent mean ± standard deviation from bootstrapping. Net ecosystem exchange (NEE) was calculated by subtracting gross primary production (GPP) from ecosystem respiration (ER): positive values are CO$_2$ emission to the atmosphere, and negative values are CO$_2$ uptake by the terrestrial ecosystem.

| Year | Group | ER | GPP | NEE |
|---|---|---|---|---|
| 2013 | D_Carex | 2.03 ± 0.10 | 3.42 ± 0.00 | -1.38 ± 0.09 |
| | | (41 ± 2) | (68 ± 0) | (-28 ± 2) |
| | D_CarexEriophorum | 1.89 ± 0.16 | 3.30 | -1.41 ± 0.24 [1] |
| | | (38 ± 3) | (66) [1] | (-28 ± 5) |
| | D_EriophorumShrub | 1.88 ± 0.53 | 4.81 | -2.93 ± 0.34 [1] |
| | | (38 ± 11) | (96) [1] | (-59 ± 7) |
| | C_CarexShrub | 2.04 ± 0.08 | 2.55 ± 0.02 | -0.51 ± 0.05 |
| | | (41 ± 2) | (51 ± 0) | (-10 ± 1) |
| | C_Eriophorum | 1.76 ± 0.05 | 3.41 ± 0.01 | -1.65 ± 0.04 |
| | | (35 ± 1) | (68 ± 0) | (-33 ± 1) |
| | C_EriophorumShrub | 2.34 ± 0.17 | 3.32 ± 0.00 | -0.98 ± 0.14 |
| | | (47 ± 3) | (66 ± 0) | (-20 ± 3) |
| 2014 | D_high (EriophorumShrub) | 3.27 ± 0.16 | 7.59 ± 0.11 | -4.31 ± 0.05 |
| | | (184 ± 9) | (404 ± 6) | (-221 ± 3) |

| | | | |
|---|---|---|---|
| D_low(CarexEriophorum) | 3.51 ± 0.19 | 5.14 ± 0.07 | -1.64 ± 0.11 |
| | (200 ± 9) | (274 ± 4) | (-74 ± 5) |
| C_high (EriophorumShrub) | 2.81 ± 0.24 | 5.85 ± 0.03 | -3.05 ± 0.21 |
| | (162 ± 14) | (312 ± 1) | (-150 ± 13) |
| C_low (CarexShrub) | 3.98 ± 0.21 | 6.20 ± 0.03 | -2.22 ± 0.18 |
| | (222 ± 12) | (331 ± 2) | (-109 ± 10) |

[1] As no bootstrapping was conducted on data for GPP, error range in NEE is only from ER.

Table 5 Root mean squared error (RMSE) and mean bias error (MBE) of the observed and interpolated fluxes (µg C-CO$_2$ m$^{-2}$ s$^{-1}$). The observed fluxes indicate those used for calibration. Period of interpolation: 22 July to 10 August (20 days); values in parentheses for 2014: 16 June to 20 August (66 days).

| Year | Group | Ecosystem respiration | | Gross primary production | |
|---|---|---|---|---|---|
| | | RMSE | MBE | RMSE | MBE |
| 2013 | D_Carex | 9 | 0.04 | 25 | -35 |
| | D_CarexEriophorum | 7 | -2 | 13 | -1 |
| | D_Eriophorumshrub | 10 | 0.05 | 26 | -29 |
| | C_CarexShrub | 8 | 0.02 | 16 | -35 |
| | C_Eriophorum | 6 | -0.01 | 17 | -27 |
| | C_EriophorumShrub | 7 | 0.002 | 14 | -34 |
| 2014 | D_high (EriophorumShrub) | 3 (7) | -0.2 (-0.04) | 32 (30) | -8 (-55) |
| | D_low (CarexEriophorum) | 8 (9) | 3 (-0.06) | 20 (22) | 3 (-54) |
| | C_high (EriophorumShrub) | 10 (10) | -4 (-0.02) | 13 (23) | 3 (-44) |
| | C_low (CarexShrub) | 12 (11) | 1 (0.2) | 23 (30) | 2 (-91) |

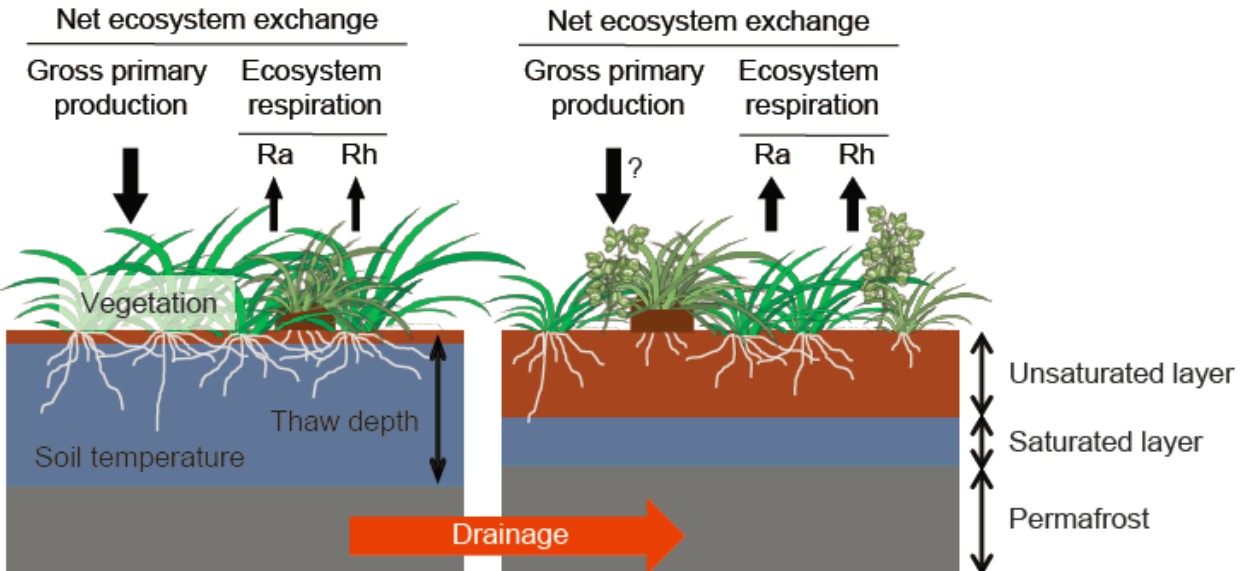

Figure 1 A schematic showing how a decade-long drainage installation affects a floodplain ecosystem and $CO_2$ fluxes. Drainage of a floodplain ecosystem alters soil temperatures through changing heat capacity and thermal conductivity, with increased soil temperatures in shallow layers, decreased soil temperatures in deeper layers, and shallower thaw depths; and decreasing the abundance of wetland grasses while increasing the abundance of shrubs. These modifications will subsequently affect $CO_2$ fluxes by changing the rates of gross primary production and possibly increasing ecosystem respiration, which consists of autotrophic and heterotrophic respiration ($R_a$ and $R_h$).

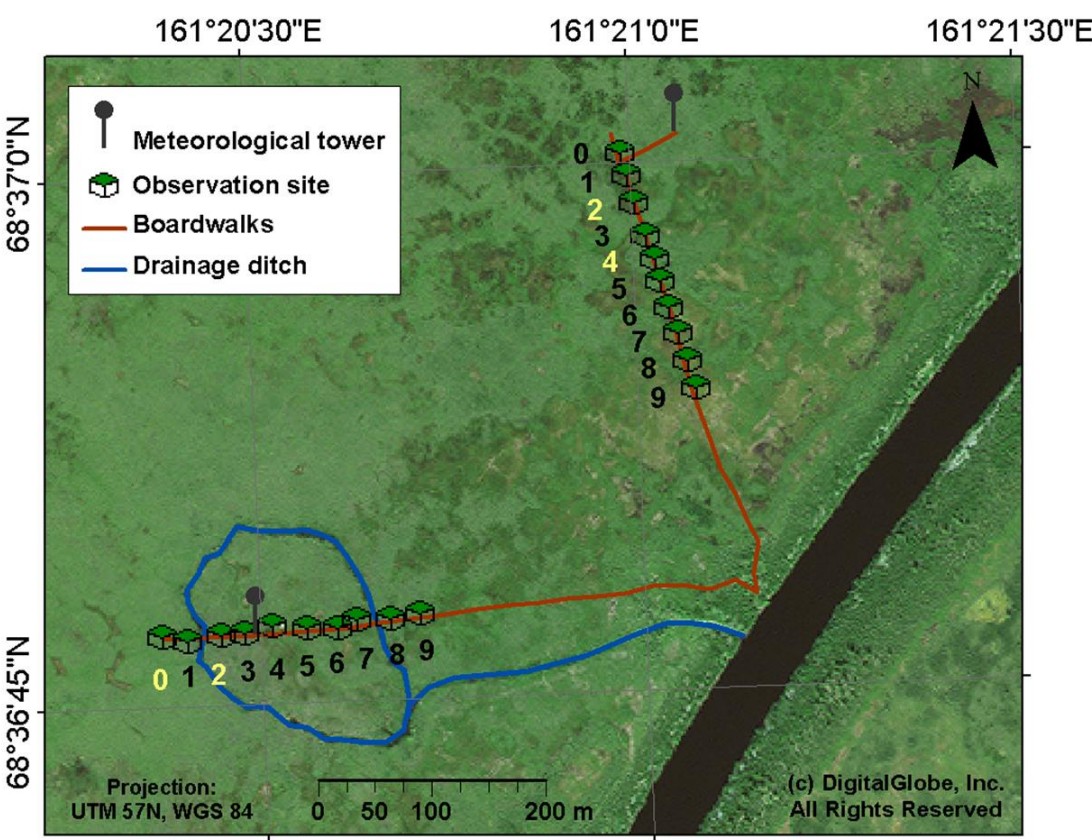


Figure 2 Aerial photograph of the site, including schematics of the drained (bottom left) and the control (top right) transects. Names of plots are written with numbers and two core plots for more frequent flux measurements are highlighted in yellow in each transect. Fluxes, vegetation community structures (created using a non-destructive method), water table depths, and thaw depths were measured in all plots, and soil temperatures were measured in

even-numbered plots only.

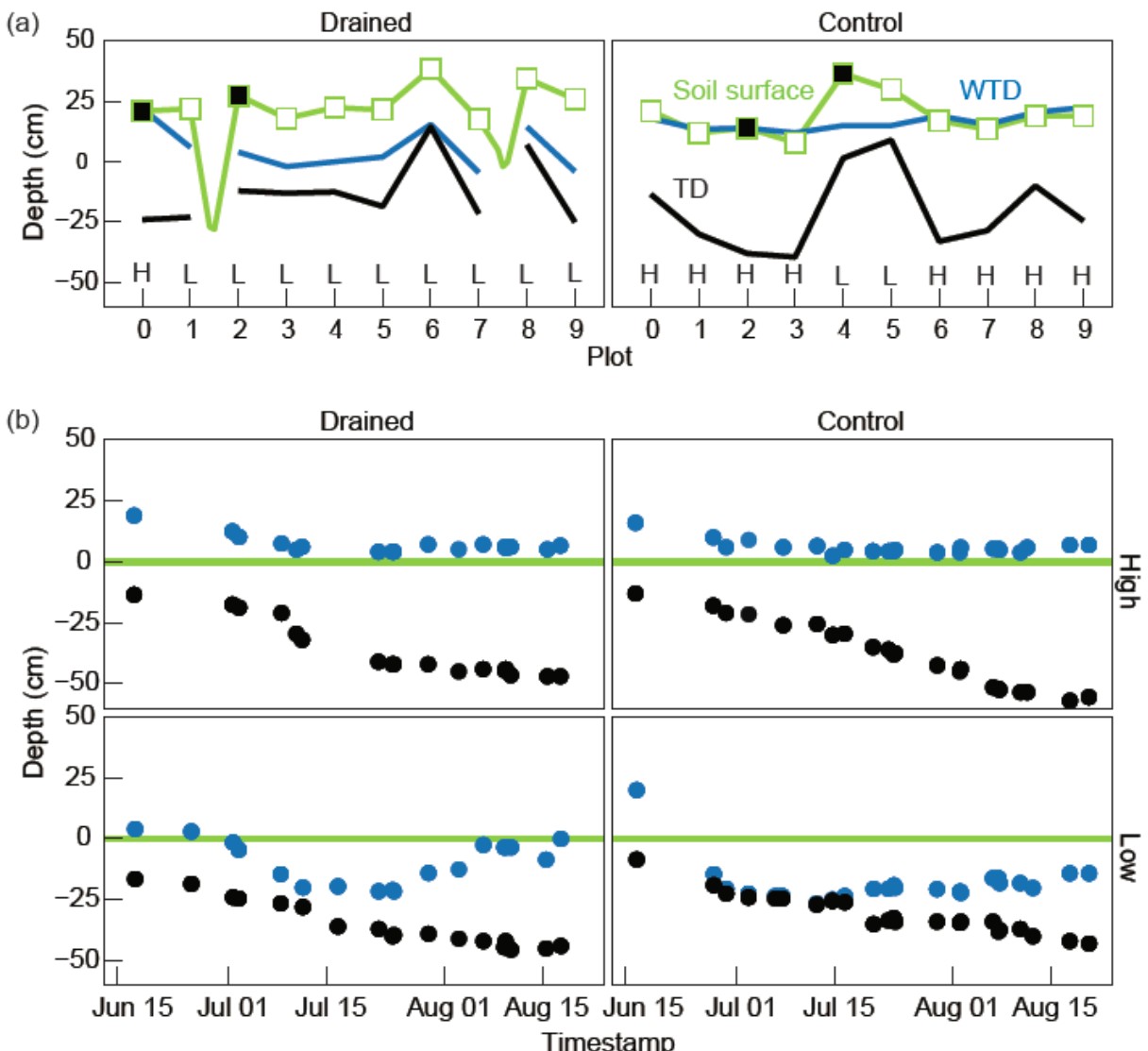

Figure 3 (a) Spatial variability in water table depths (WTD, blue lines) and thaw depths (TD, black lines) measured across the two transects on 10 August, 2013. Green lines indicate terrain height, with plots indicated with squares (core plots = closed squares). The letters H and L indicate the high and low WTD category of each plot, respectively. (b) Temporal variability in WTD (blue points) and TD (black points) observed at the four core

plots over the growing season of 2014, separated by transect (columns) and WTD category (rows). Green lines represent soil surface.

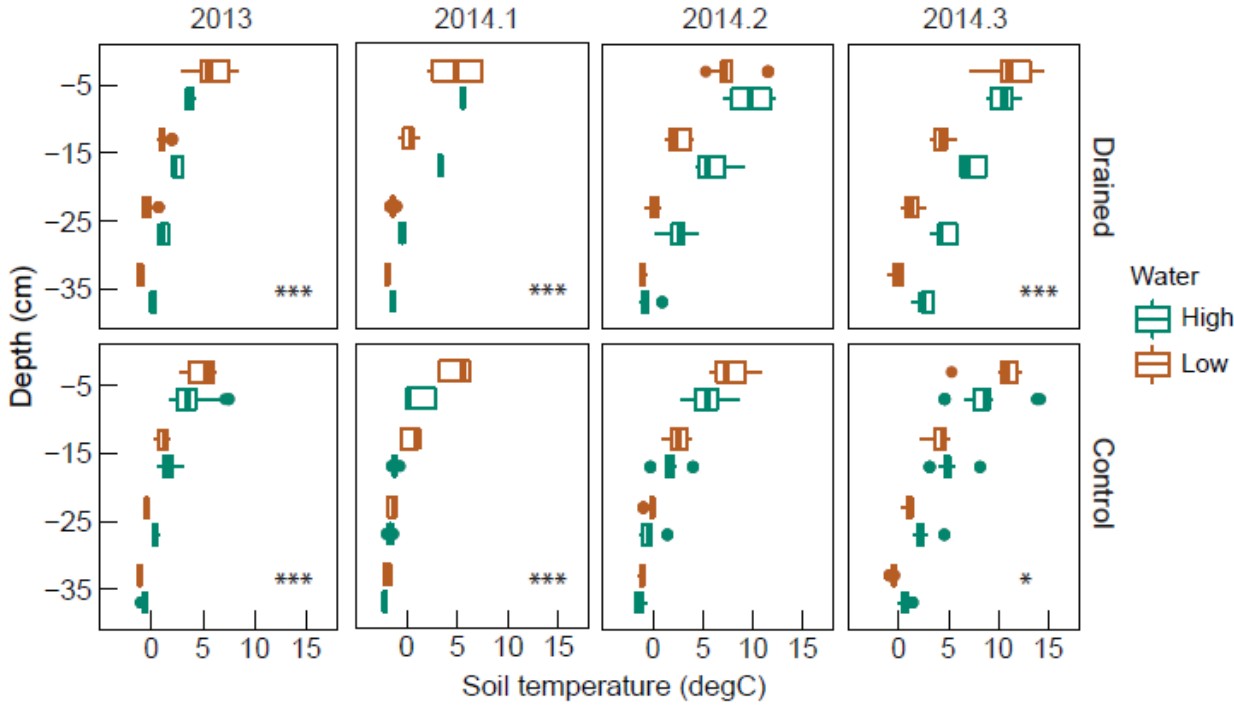

Figure 4 Soil temperature profiles based on observations at 5, 15, 25, and 35 cm depths from even-numbered plots each in the drained (top) and control (bottom) transects. Boxplot contains median, 25 % and 75 % quartiles, and ± 1.5 interquartile ranges. To minimize the impact of the diurnal temperature cycle on this temporally discontinuous dataset, the time window for averaging was restricted to 1 to 5 pm. Panels from left to right show data from 2013, as well as from three sub-seasons of the growing season of 2014: (2014.1) 15 June–5 July,

(2014.2) 6 July–26 July and (2014.3) 27 July–20 August. Green color indicates plots with high water table depths (WTD) and brown indicates those with low WTD. Data subsets where significant differences in WTD between the high- and the low-WTD plots were detected are marked with asterisks ($P$ value $< 0.001$ ***, $< 0.01$ **, $< 0.05$ *).

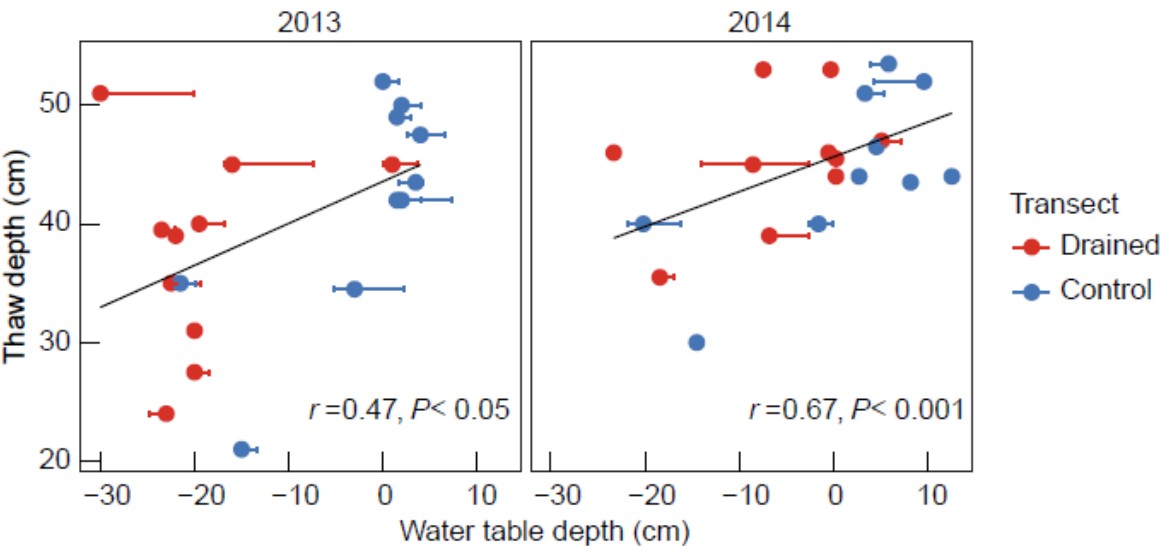

Figure 5 Correlations of thaw depths (TD) and water table depths (WTD) in mid-August 2013 and 2014, where

red points indicate plots from the drained transect and, blue points plots from the control transect. Error bars of

WTD represent the minimum and the maximum ranges of WTD of the previous 20 days. Results of correlation

analysis for each year are presented with black lines.

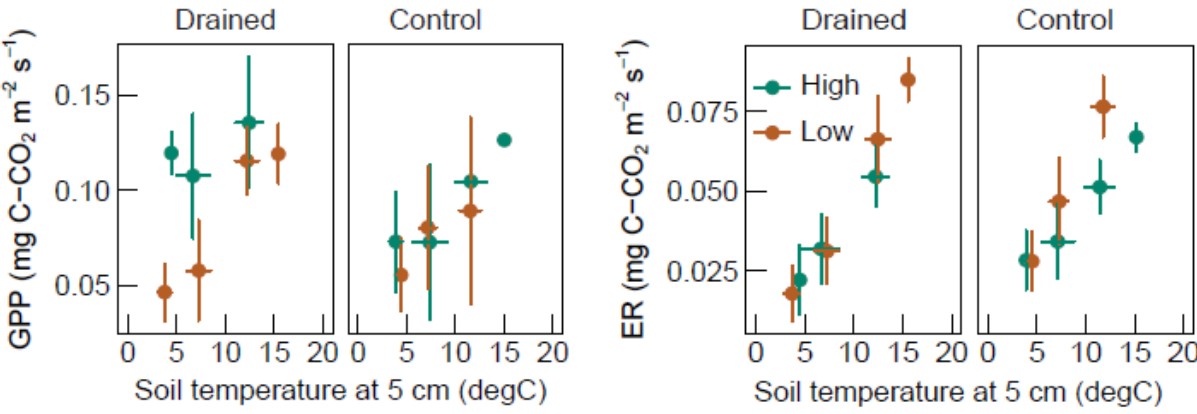

Figure 6 Links between average soil temperatures at 5 cm and GPP and ER rates. Data are from 2013 (20 July–10 August) and sub-season 2014.3 (27 July–20 August); both cover similar phenological periods. Data were grouped into temperature bins of 5 ℃.

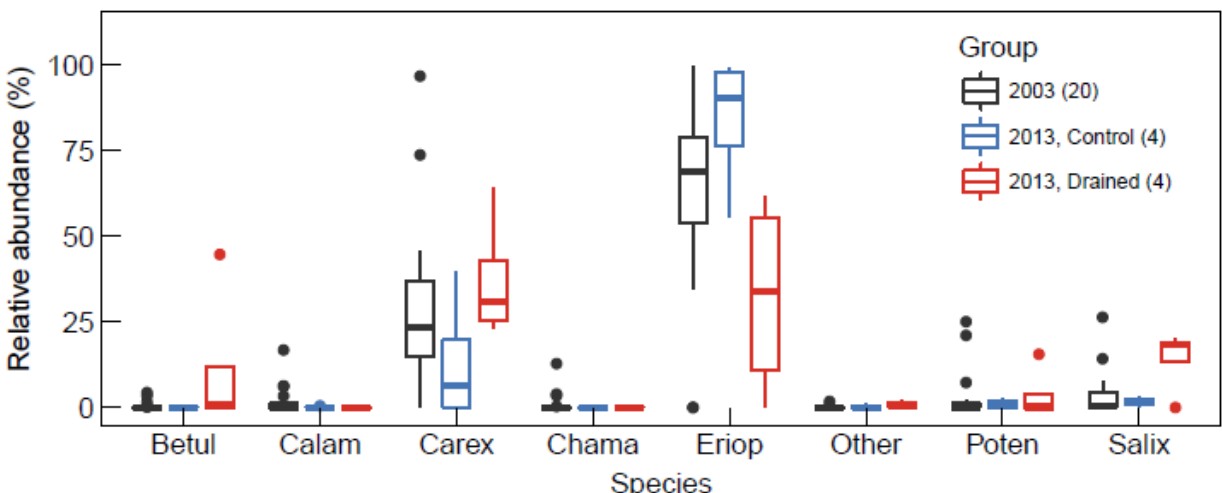

Figure 7 Abundances of vegetation species observed across the transects in 2003 and 2013. Numbers in parentheses are the number of replicates. Boxplot contains median, 25 % and 75 % quartiles, and ± 1.5 interquartile ranges. Betul: *Betula exilis*, Calam: *Calamagrostis purpurascens*, Carex: *Carex* species, Chama: *Chamaedaphne calyculata*, Eriop: *Eriophorum angustifolium*, Poten: *Potentila palustris*, Salix: *Salix species*.

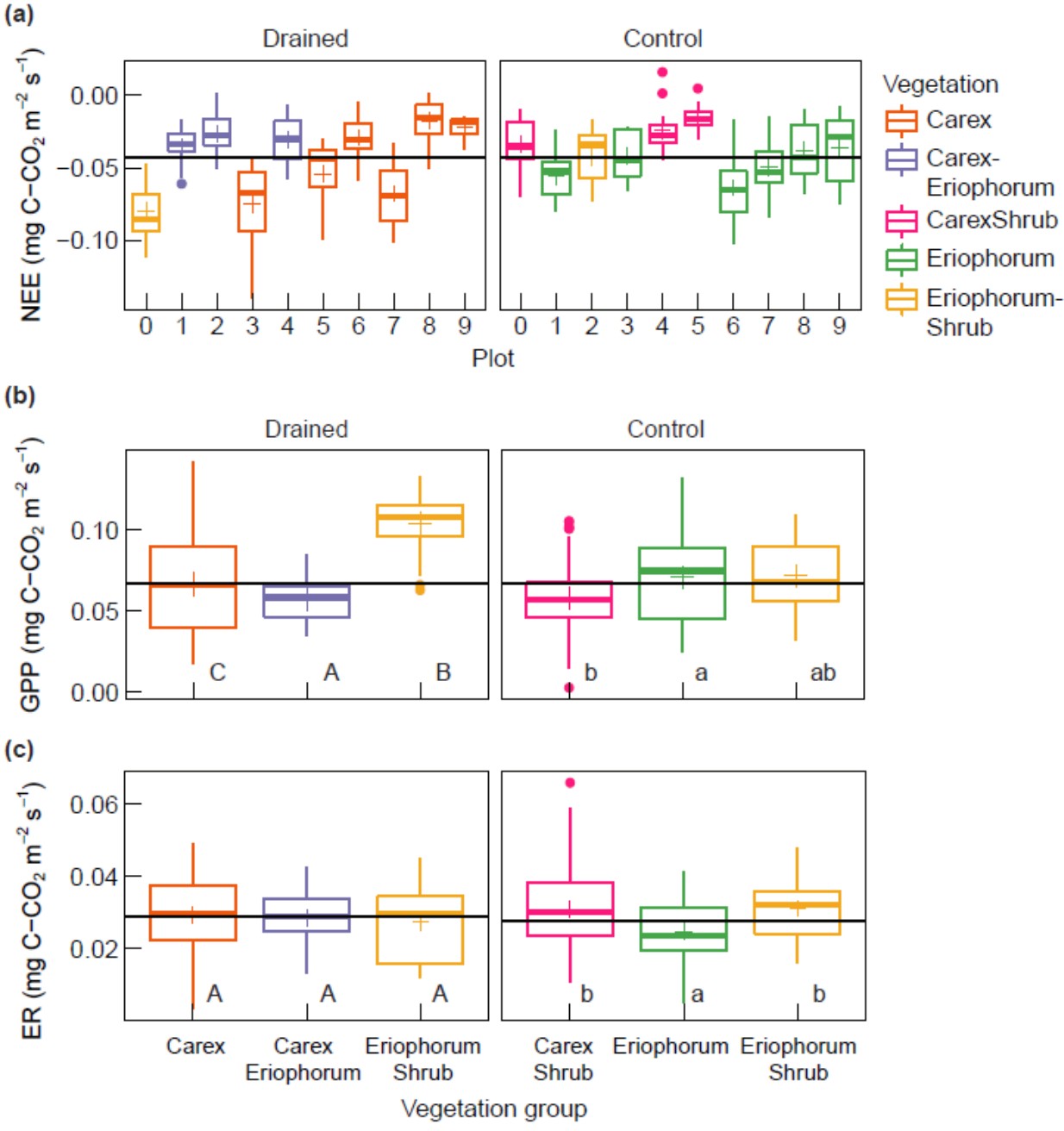

Figure 8 (a) Variability of net ecosystem exchange (NEE) among individual plots during the 2013 growing

season. Boxplot contains median, 25 % and 75 % quartiles, ± 1.5 interquartile ranges, as well as mean values

with cross points per plot, with colors indicating the dominant vegetation species. The black horizontal bars show the mean flux rates averaged for the entire transect. (b) Gross primary production (GPP) and (c) ecosystem respiration (ER) rates aggregated by vegetation group. Significance of differences between groups, determined by one-way ANOVA and Tukey's post hoc test, is indicated by the letters. Different letters indicate significant differences between groups while the same letters indicate significant similarities.

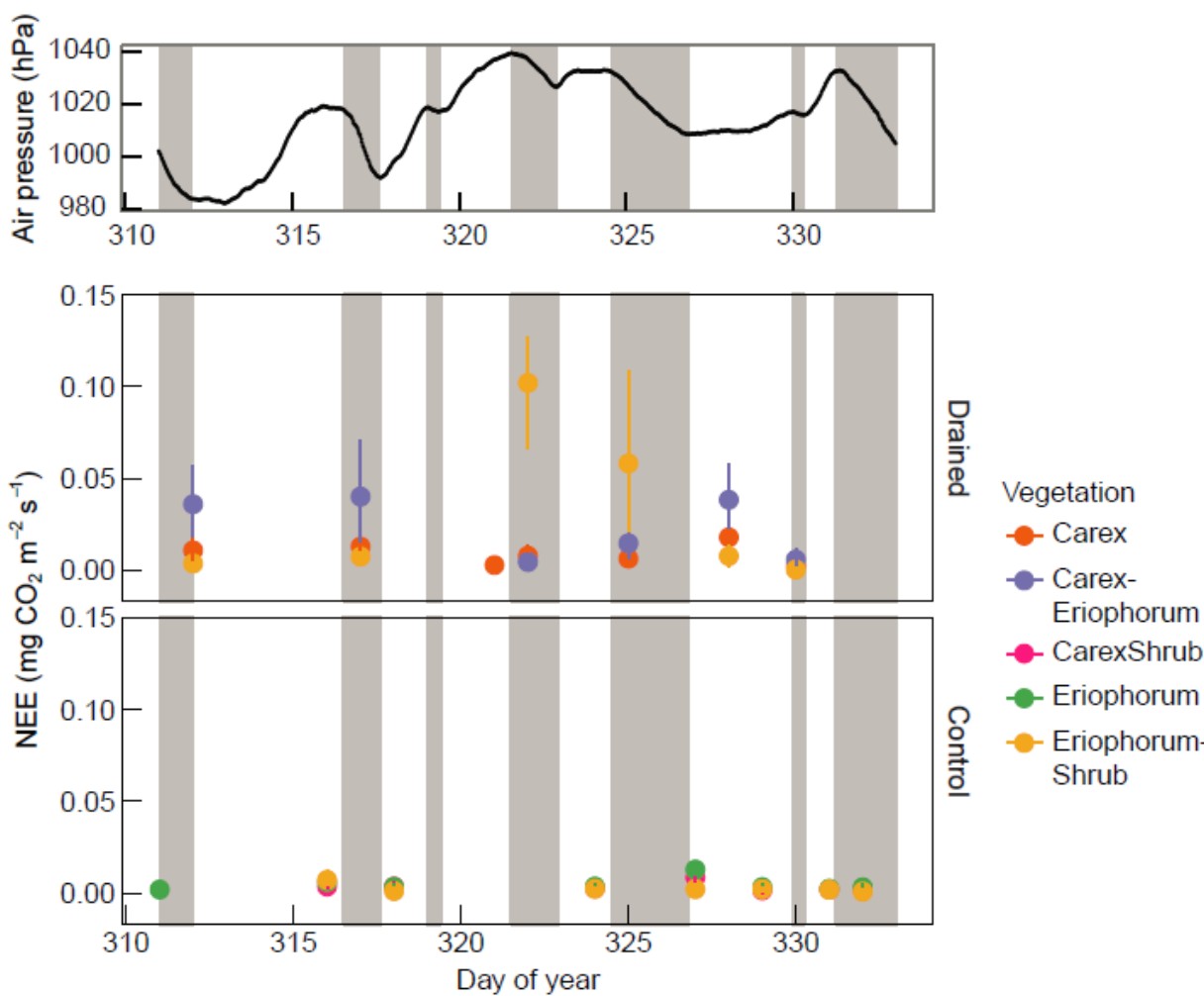

Figure 9 Atmospheric pressure (top) and net ecosystem exchange (NEE) from November 2013 by vegetation type (color), presented in parallel. Time periods with decreasing air pressure are shaded in gray. NEE fluxes were pooled by vegetation group (color) and mean values and standard deviations were plotted.

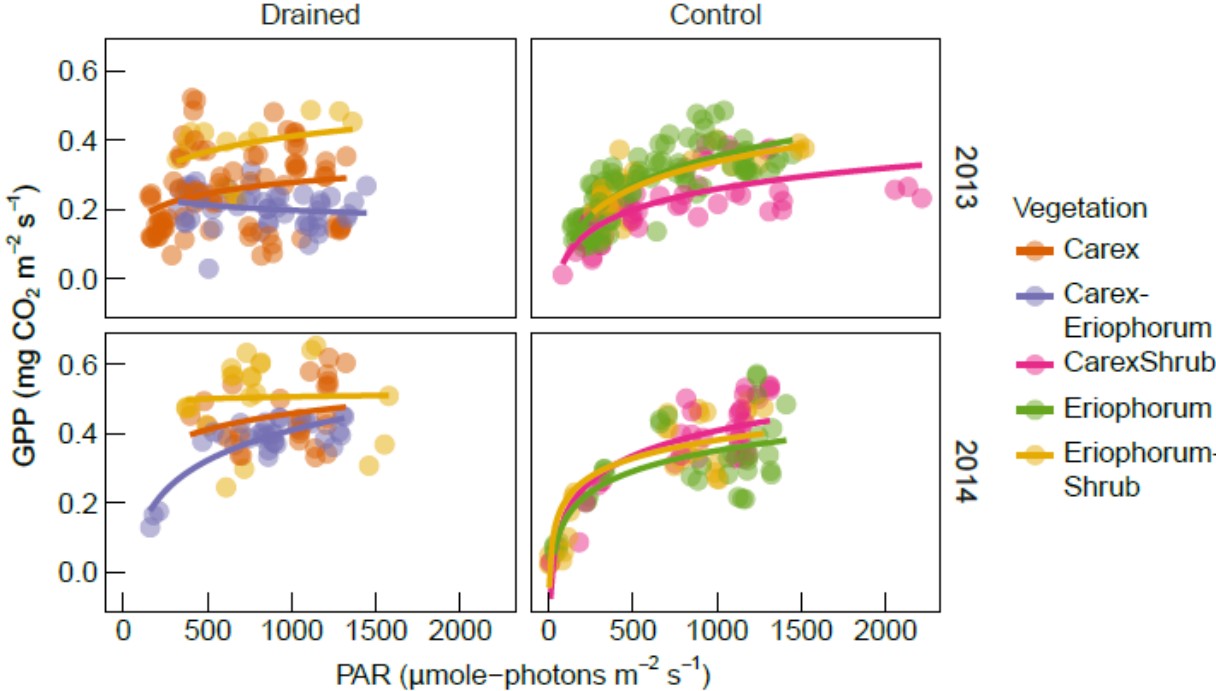

Supplementary Figure S1 Logarithmic relations between photosynthetically active radiation (PAR, X axis) and

935 gross primary production (GPP, Y axis) by transect (columns), vegetation type (color) and year (rows). Data

points are only from August when vegetation activity was high enough to minimize seasonality.

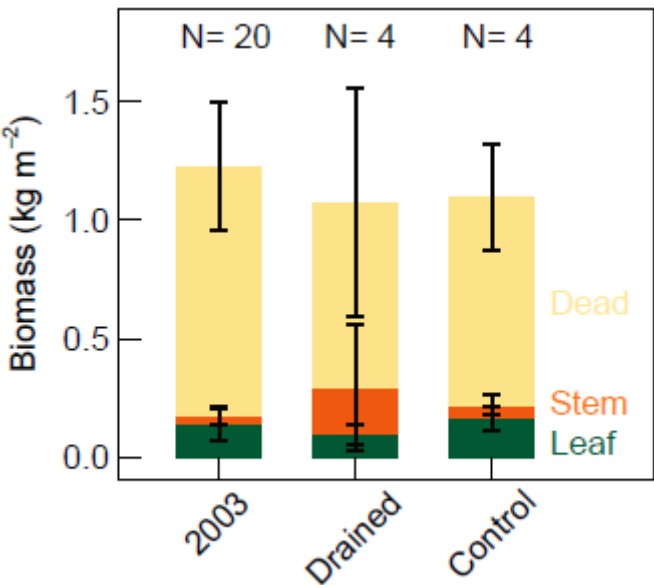

Supplementary Figure S2 The total aboveground dry biomass (mean ± standard deviation) of standing dead and living plants, measured in 2003 ($N = 20$) and 2013 ($N = 4$). Weights of dry biomass were not separated by species. Dead: standing dead materials, largely from *Carex* species and *Eriophorum angustifolium*; stem: stems, mainly from shrub species; leaf: green leaves of all species.