# Peer review of "Long-term drainage reduces CO2 uptake and increases CO2 emission on a Siberian floodplain due to shifts in vegetation community and soil thermal characteristics"

_Biogeosciences, 2015_

## Referee Comment (RC1) · Anonymous Referee #1 · 24 Feb 2016

The manuscript by Kwon et al. reports on two years plus some historical data of carbon dioxide flux measurements made in a tundra wetland in Siberia. Thereby, the authors compare a natural wetland with a drained site, which are in near proximity of eachother. Such "paired" sites are rare and strongly needed to further understand carbon dioxide exchange of tundra ecosystems. Moreover such experimental manipulations as shown in this study are rare and difficult to setup in remote regions, while being of extraordinary importance to understand ecosystem functioning under ongoing climate change. Therefore the represented manuscript is of great interest for the readers of Biogeosciences. Unfortunately, the authors present carbon dioxide fluxes only,

whereas methane fluxes are likely to contribute considerably to the total carbon budget of this type of ecosystems and methane can easily become the game changer as already suggested in previous study at the site a decade ago. Still, the analysis based on the vegetation mixtures as well as ER, GPP and NEE provides a valuable contribution which has not been covered in previous studies. Besides the currently presented science and given the fact that the author list includes native English speakers and Senior scientists it seems like not all co-authors have read the manuscript or provided input. If done so the readability of the paper can clearly be improved and would fulfill minimum scientific standards – especially in the discussion section. This must be done in order to have the manuscript considered for actual publication in any journal.

Please find other major comments and a list minor/technical comments, which should be taken care of by the authors prior to possible publication in Biogeosciences.

Major Comments: (1) The manuscript structure needs to be improved in terms of avoiding mix-ups between site, location, transect as well as why certain places were characterized by high WTD even though being located in a drained area. You need to explain why such patterns occur and why you decided to separate these. (2) Furthermore when you investigate component fluxes it remains unclear why the authors once use the vegetation composition as main driver variables why for ER only WTD was choses as primary driving variable. Since both component fluxes are interlinked as well as species composition depends on WTD I suggest to find a common ground, in other words combinations of species mixtures and WTD and then look at the individual fluxes. (3) It remains puzzling why the other present 20 and 66day cumulative CO2 fluxes in the manuscript. Such cumulative numbers are for two reason not comparable to other sites: (i) providing a mean or a cumulative average for such a period in mole and (ii) the integration period is not the same for the two year and does neither represent a year or a specific season. I suggest instead of presenting cumulative sums to clearly look at the processes and driving factors since there seems to be quite some potential in the dataset to do so. (4) The discussion needs substantial improvement

concerning structure, scientific argumentation as well as concerning the logic. There are few reference but too often restating of results.

Minor /Technical Comments:

P2,L27: I think the final sentence is not clearly representing your results. You have both a Pro and a Con result so far from the 2013 and 2014 data. Why would this lead to the current conclusion? P2,L33: until which soil depth and what is your definition of the Arctic? P2,L41: delete "that is currently stored in deeper" P2,L42: please try to be specific: Arctic ecosystems or ecosystems in the Arctic. The Arctic is not an ecosystem you may refer to it as a biome or geographical zone P2,L44: What about radiation and nutrient supply – both crucial factors in the Arctic too – especially since you mention PAR in your figures P3,L49: why complex? Indeed this may lead to changes but what makes it so complex, try to be more specific P3,L52: One could add an objective here again, otherwise this is all nice information but the reader does not yet know why this background is important P5,L91: Which conclusion do you draw from your introduction? My suggestion would be to state the need for more studies needed, paired design studies, extrapolation to models according to x,y,z P5,L92ff: this is important but seems a bit lost here - can you incorporate this information at the beginning of the previous paragraph? P5,L97: depends on how you define short-term – the historic data refers to four growing seasons of measurements including one experimental season. How about " an initial hydrological manipulation a decade ago" since you have two season only and can hardly use this as a long-term study. P5,L98: "This study investigates..." P5,L104: There hasn't been much on frozen season information been provided in the introduction but one of your foci is particularly on frozen season fluxes. Therefore I suggest to expand towards this topic a bit.

P6,L112: you state "annual mean" – for which year or which timeperiod is this provided P6,L115: According to the previous papers on this site, the spring flood occurs occasionally. Was this the case for both years of observations? P6,L115: You refer to water table depth even though the water table is above the soil – it's a bit contradicting since

this does not refer to a depth but rather to a height P6,L116: To which period are you exactly referring to with "early growing season"? Can you specify this – see also the comment to Figure 5 P6,L117ff: please rephrase: Periodic fluctuations approx.. every 5 years but no persistent long-term in air temperature and precipitation could be identified when investigating meteorological data since 1980. P6,L120: remove the "of" before 2004 P6,L120: can you please clarify what you mean with " water from the surrounding are". Surrounding what? And what were the effects was only the inner area of the drainage ditch as seen in Figure 2 drained or also the outer area? P6,L123: how do such short term fluctuation occur? P6,L125: this is an objective and a statement that has been given before – please avoid redundancies and move objective from the methodology section to the introduction P6,L127: is that the previously drained site or an new drained site? P6,L129: How did you estimate that this area was not affected? P6,L129: try to avoid a mix up of the term site and plot/sampling location. I assume you are referring to plots at the two subsites or the one site in Chersky, depending on the fact whether you have seperate experimental fields for instance or other criteria P7,L133: How do you define representative vegetation? Representative for what? P7,L133: "small enough. . .." – does that bias your measurements/results? P7,L134: I suggest to rephrase the sentence towards: "Our analysis comprises the analyis of x weeks of y during .... P7,L136: you may delete the full sentence P7,L138: repharse to: Subsites and plots are labeled according to. . .. P7,L145: What about soil temperature and soil water measurements in the chamber as well as frozen or not frozen status? P7, L145: Did you test the chambers to avoid Venturi effects? P7,L152: which should then equal ER, correct? To my opinion is should be dark during the frozen time most of the time anyway P7,L155ff: very good and I can imagine how much work this must have been in this region P7,L159: please correct throughout the whole manuscript: $\mu$mol m-2 s-1 instead of $\mu$mole P8,L159: I suggest to add a citation of a typical chamber flux calculation paper P8,L169: what do you mean with nearby location? P8,L172ff: Why on dry biomass, if you have very "fleshy" plants they could have a larger realistic contribution to the overall biomass then drier plants but if you base this on dryweight

only, the effect might disappear P9,L179ff: this is unclear. also, how do the two different methods compare? P9,L182: Write full words in the heading P9,L187: data are always plural P9,L189ff: phrasing is incorrect – I assume that the coauthors that are native speaker could solve this issue. Just as a side note, one could easily reject a paper due to such "technical errors". At the moment it is rather an obvious sign that not all co-authors have fully read the manuscript. P10,L204: why does this approach make it better to compare with 2013 data? P10,L205: can you name these pre, peak, post season and can you do this similar in all papers originating from this research P10,L207: why august? P10,L210: Please replace "to find out" with "To investigate" P11,L218ff: This is very confusing, please simplify P11,L223: what do you mean with -1 . GPP? P11,L236: . . .from sensors installed in "the" chamber system. . . P12,L239: how did you optimize, via bootstrapping methods? P12,L247: please remove "equation" P12,L249: measurements P12,L251:. . .. temperature from the meteorological station P12,L254: insert "," between 2104 and while P12,L254: numbers below eleven are commonly given in words P12,L255: please change to Carex sp., Erophorum angustifolium. . .. P13,L266: wording P13,L267ff: name the other terms P13,L269: How did you add both error term? P13,L270ff: how many and why? P13,L274: time statement goes usually to the end of the sentence P13,L275: please explain since these are two antagonistic fluxes. were both fluxes low or large but equal? P13,L275: please refer to Figure 3 here and for Figure 3 I suggest to state very clearly that one panel is based on 2013 data only and one is 2014 data only! P13,L279: . . .2014. However... P13,L280: but this happened at both sites? P15,L304ff: please simplify. I also wonder why you treat vegetation structure independent from WTD and TD even though these are obviously linked P15,L312: That is something that could be indicated in your Figure 1 scheme and make the whole story much clearer P15,L318: well done, but why do you give r values and its sufficient to show the numbers either in the figure or the text but there is no need to repeat these P15,L321ff: Fine, but what does this tell us? P16,L326: you might want to consider stating this slightly different-take the maximum uptake as 100% and then just give with > CarexShrub (how many of 100% remained) would there be a

difference between CarexEriophorum and EriophorumCarex? Meaning does the order of the names mean that the first was more abundant? your abundance was based on the 10% criterion, so there were no plots with all three plant species? P16,L329: here I suggest a bar diagram, so that the reader is able to follow. P16,L332: I suggest to not jump between drained and undrained an veg. groups - one after another P16,L339ff: This is redundant P16: I suggest to find a way, to simplify the drained undrained and species mixtures.... see also the comments given to the figures P16,L343: soil temperature effects of what? P17,L344: why do you explain one Variable with vegetation primarily and the other component with WTD even though both are linked? P17,L349: Table, Figure? P17,L355ff: I have three general comments here. I think its better to focus on the actual results instead of doing an error analysis, discussion first – this can be done either in the methodology or in the discussion. Secondly you do not have too much data to do a robust gap-filling and base results on the gap-filled data, therefore I suggest to stick with the measurements primarily. Also you provide cumulative flux values for 20 days in 2013 and 66 days in 2014. How relevant are these or how do you suggest one should set this in perspective to other studies? My suggestion would be to remove this part or if this is not possible to focus on specific season and also keep same time intervals in both years. Last but not least cumulative fluxes should be given in g C or CO2 m-2 per time interval not in $\mu$mol to allow comparison. P18,L376: PAR was excluded.... P18,L373ff: if the model considered vegetation type then of course you will find the same effect as before. Please clarify P18,L378: 0.3$\mu$mol m-2 20days – that's basically nothing and your overall uncertainty of the approach is larger. P18,L381: how can these be similar- can you calculate the fluxes also for a time period of 20 days? so that they become comparable to 2013? please give g C since $\mu$mol do not make sense for cumulative fluxes Furthermore I wonder what the key message is if you present 20day or 66day cumulative fluxes. How shall these be compared to other studies, annual or growing season estimates would be much more relevant. P18,L385: When were these observed? P19,L387: replace "by" with "of" P19,L387: rephrase to "emitted on average four times more CO2 than the undrained site." P19,L393: Can

you authors state why this is the case? P19,L397: how large was the variability, if its too narrow anyway then why would one test it? P19,L398: I think you could provide this amount of information in a brief table or add it do an existing table. P19,L406ff: the mixing of the two sites and the individual sub-locations which are either wet or dry is still very confusing and I suggest to clearly explain this in the beginning but also make this always clear for the whole manuscript. P20,L409: What is "low WTD of undrained_low..."? P20,L411: Which do you think a location with a low water table in the drained area is quickly affected by precipitation? Isn't that contradictory to your experimental setup where you have a drainage channel to actually drain the water? P20,L412: Do you have a reference proofing that Eriophorum is capable of achieving such water holding? P20,L414ff: this is unclear, please explain and clarify P20,L417: please delete this paragraph. P20,L421: Here you state it's a clear vegetation effect but I suggest to argument differently. You introduce a disturbance such as drainage and this has a follow-up effect e.g. on vegetation and then subsequently the carbon fluxes are affected. Please make sure you have a clear logic in your manuscript. P20,L423: replace "died out" with "extinct" P20,L425: That is a very precise result and I wonder how you can determine this so accurately. The following sentence contain the same information, please avoid redundancies. "... with larger shrub abundance in the future." P21,L435: Appendix 3 – consider that some of your figures will be available in b&w only.The currenty color choice makes it impossible to see the error bars P21,L436: "... was a reduced abundance of..." P21,L438: Please explain why a site that contains Eriophorum will have an even larger decrease in GPP compared to the Carex while you state before that Eriophorum is so productive compared to Carex. This is contradictory. P21,L440ff: Is there any proof for your hypothesis? References are fully missing? Also try to avoid restating results P22,L451: prevalence -> occurrence P22,L455: Can you estimate how much lower the CO2 flux could be? What is the picture of your results? P22,L459: That is the first time you mentioned Rh specifically, Why were there no specific results on this part presented in the results section? P22,L462: soil surface temperatures instead of surface soil temperatures P23,L481: Where did you find the

Rh increase in your results?

Figure 1: Even though the authors do a great job in bringing up such scheme about the experiment it seems incomplete. What are the fluxes before drainage and I suggest to modify the size of the arrows to see what you hypothesize. Other effects such as vegetation effects are not visible here. The Figure caption refers to a drainage event. When reading the manuscript this seems not to be an event. I also suggest to use the same terms as used in the text for photosynthesis and ecosystem respiration (GPP, ER) and to include NEE.

Figure 2: this is not a schematic but an aerial photograph. How about: Aerial photograph of the site with the schematics of the drained and undrained transects. Names of observation locations are indicated with numbers and the core locations are highlighted in yellow. - I further suggest to explain what core locations are and what was observed at the sampling locations briefly since the figure should be fully self-explanatory without the manuscript text

Figure 3: I suggest to highlight the year for the various panels as well as to indicate the wet and dry locations in the respective transects, since this particular issue may lead to lots of confusion. What are "relative terrain heights"?

Figure 4: Why not providing common boxplots? Figure caption: Abundance of …. Betula exilis, Eriophorum angustifolium etc…

Figure 5: I suggest to the subseasons by name instead of 2014.1 etc.. Why did you choose the exponential interpolation approach? Also name the subseasons in the Figure caption

Figure 6: this is very well explained in the text and I suggest to bring this issue up at an earlier point in the manuscript, since this proofs the concept of your experiment and by these structural changes an influence on the $CO_2$ fluxes becomes relevant.

Figure 7:I suggest boxplots instead of the clouds

Figure 8: Consistently explain the length and a name for the subseason throughout the manuscript

---

## Referee Comment (RC2) · Anonymous Referee #2 · 4 Mar 2016

The topic of the submission is undoubtedly of interest to the readership of Biogeosciences Discussions as it presents some interesting data arising from an experimental manipulation and its impacts upon ecosystem functioning plus gas exchange in a floodplain. Of particular note is the attention to phenomena during the frozen season as well as the thaw season. However, my greatest criticism of the overall project is why investigate CO$_2$ exchange in such temporal and spatial detail and yet completely ignore methane, and for that matter nitrous oxide exchange and changes both spatially and temporally. It is surely the balance of changes between these contrasting greenhouse gases, of differing radiative forcing strength and atmospheric concentrations that

is key here? I miss this vital context entirely within the paper as currently submitted. The abstract is of a suitable length and is reasonably informative, but I do question the concluding remark. It is not sufficiently explanative in relation to the findings of the experiment. The remainder of the text is rather lengthy and could be substantially consolidated without losing any impact (in fact making it much more impactful). The number of figures is verging on excessive. They could be perhaps consolidated and some relegated to appendix/supplementary materials, providing the key focus the manuscript currently lacks? Page 2 line 25 onwards (and subsequent incidences) -the use and comparison of cumulative figures for 20 days in year 1 and 66 days in year 2 does not seem to me to be sufficiently clear to the reader. The premise and importance of the study are generally both well explained within the introduction section of the manuscript. Line 114: 'Reliable Prognosis"? What is this – it requires more careful explanation. Line 118: what is the magnitude of such fluctuations? Line 120: the depth of the drainage ditch was? Line 129 'affected' not 'effected'. How do you know that the drainage ditch had no effect at 600 m away – how did you ascertain this? Line 135: It is not clear to the reader what the rationale was for the 3 weeks sampling in 2013 and then 10 weeks in 2014. This should be made clear. Line 136: you introduce the term transects but have not done so before m- this is very confusing and should be addressed fully. Likewise, the labelling of transects is poorly defined. Line 141: The PVC collar was installed permanently, but how so and when, and to what depth was the soil isolated? For how long was the collar installed prior to sampling beginning? i.e. how long was there for recovery of the vegetation? Did cutting-in of the collar lead, as in many cases reported elsewhere, and in my own considerable experience, lead to vegetation damage/death in any case? Line 154: What evidence do you have that using ice packs effectively worked to keep the temperature constant as you claim? Line 158: the units quoted need attention. Line 165 onwards: Your phrasing is not sufficiently clear here regarding the 'conflict' between choosing core sites based on WTD category Line 170: define and quantify 'nearby' accurately please. Line 174: It is not clear how the data from 2003 form a reference! Lines 175-181: I find the explanation

here insufficiently clear. You need to provide the reader with a much clearer explanation of what you did and why! Lines 188-189: Aerobic incubation of soils – insufficient detail is provided here for this to be repeated. Lines 206-207: There is insufficient detail regarding why you chose August as your point of reference. Line 223: You do not describe the term -1. GPP. This should be corrected. Line 269-70: how exactly did you add this error range in each case? Section 3.3.1. I find this, as currently written, to be overly complicated in structure and terminology. The authors need to simplify their scheme substantially and really pull-out the key take-home points from the data. A key issue for me in this respect is that you are comparing data with different time periods of coverage between years. Surely a direct comparison between same time periods each year would be more helpful to the reader? In addition, your choice of use of cumulative data seems at odds with your choice of units. This requires correction. Line 396 onwards: low variability of what exactly? In the discussion section, I find the terminology again confusing – clearing this up substantially earlier-on in the paper and then following this through to the discussion would really help the reader comprehend the key messages much better than at present! Line 428: this is rather a weak initial statement – be more specific. Line 440: What direct evidence do you have to back-up this clear statement? Figures: (1) A useful conceptual figure, but much better reproduction is required. There seems to be a mis-match between continued drainage in the experiment and 'events' stated in the legend. This must be addressed. Figure legend terms do not match directly those used in the text of the manuscript. (2) You need to include the measurements taken at each observation site. (3) OK (4) Grey points are not sufficiently clear on this figure. SD error bars – n = ? (5) Sub-seasons of 2014 are unclear – see my comments on this issue in the text of the manuscript also. (6) OK – but why is this Figure 6 – surely this information warrants being at the start of the explanations!! (7) Clouds of grey points are hard to distinguish – rethink...such as boxplots for example. These would, I think be much clearer for the reader. (8) Ditto (9) OK (10) OK (11) Comments as (7) Appendix figures OK. There are numerous points in the text where small edits of the correct word are required for clarity of the narrative.

There are too many to list here, but a native English speaker should be consulted to address these shortfalls. This is the worst such manuscript in this respect that I have read in some years.

---

## Author Comment (AC1) · 24 Mar 2016

*Comments from the reviewers are in black and responses from the authors are in blue. Responses from the authors are sometimes in the past tense because the authors have already been editing soon after receiving comments from the reviewers.*

**Authors' responses to the Reviewer #1**

The manuscript by Kwon et al. reports on two years plus some historical data of carbon dioxide flux measurements made in a tundra wetland in Siberia. Thereby, the authors compare a natural wetland with a drained site, which are in near proximity of each other. Such "paired" sites are rare and strongly needed to further understand carbon dioxide exchange of tundra ecosystems. Moreover such experimental manipulations as shown in this study are rare and difficult to setup in remote regions, while being of extraordinary importance to understand ecosystem functioning under ongoing climate change. Therefore the represented manuscript is of great interest for the readers of Biogeosciences.

Unfortunately, the authors present carbon dioxide fluxes only, whereas methane fluxes are likely to contribute considerably to the total carbon budget of this type of ecosystems and methane can easily become the game changer as already suggested in previous study at the site a decade ago.

Author comment: We fully agree with the reviewer that changes in the methane budget significantly contribute to the net effect of the drainage on the carbon cycle processes within this ecosystem. However, considering the detailed effects of changes in the water regime related to the methane emissions, which include e.g. shifts in microbial community structure or methane transport pathways such as ebullition and plant mediated transport, combining $CO_2$ and methane fluxes would result in too much material for a single manuscript. We therefore decided to present the analysis of drainage effects on methane fluxes in a separate publication, and exclusively focus on $CO_2$ fluxes within this manuscript.

Still, the analysis based on the vegetation mixtures as well as ER, GPP and NEE provides a valuable contribution which has not been covered in previous studies. Besides the currently presented science and given the fact that the author list includes native English speakers and Senior scientists it seems like not all co-the authors have read the manuscript or provided input. If done so the readability of the paper can clearly be improved and would fulfill minimum scientific standards – especially in the discussion section. This must be done in order to have the manuscript considered for actual publication in any journal.

Author comment: All authors listed for this manuscript were involved in discussing the presented material at regular intervals, therefore also native speakers contributed to proofreading and editing the submitted manuscript. However, we do not think that it is the role of the native speakers on the author list to take care of a word-by-word text editing. Instead their focus was placed on the scientific content and the overall readability and understandability of the manuscript. Of course we also do not want to burden the reviewers with language editing, but we were relying on the text editing services offered by Copernicus journals after completion of the review process to straighten out flaws in the language. Since the reviewer's comments imply that the quality of the language impairs the scientific message of this manuscript, we will have the revised manuscript version checked by a professional language editor before re-submission.

Please find other major comments and a list minor/technical comments, which should be taken care of by the authors prior to possible publication in Biogeosciences.

Major Comments:

(1) The manuscript structure needs to be improved in terms of avoiding mix-ups between site, location, transect as well as why certain places were characterized by high WTD even though being located in a drained area. You need to explain why such patterns occur and why you decided to separate these.

The terms site/location/transect were all corrected throughout the manuscript to avoid confusion. Also, the authors added some sentences to give more explanation why high- and low-WTD plots existed and were divided.

(2) Furthermore when you investigate component fluxes it remains unclear why the authors once use the vegetation composition as main driver variables why for ER only WTD was choses as primary driving variable. Since both component fluxes are interlinked as well as species composition depends on WTD I suggest to find a common ground, in other words combinations of species mixtures and WTD and then look at the individual fluxes.

There seems to be misunderstanding. The authors tried to show that "drainage changed vegetation community structure and soil temperature regime, and then these two modifications altered $CO_2$ fluxes". The authors linked shifts in these two environmental factors with patterns in $CO_2$ fluxes, but there was no part where we tried to explain ER solely by WTD. If the reviewer refers with this comment to P17, L344 the detailed answer is written below. To summarize, the authors divided datasets into high- and low-WTD plots and investigated the relationships between soil temperatures and ER. In this way, soil temperature effects on ER were investigated separately for different WTD categories, but we did not evaluate direct WTD effects on ER. Nonetheless, the authors edited parts of the text where such relationships may have been presented in a confusing way.

(3) It remains puzzling why the other present 20 and 66 day cumulative CO2 fluxes in the manuscript. Such cumulative numbers are for two reason not comparable to other sites: (i) providing a mean or a cumulative average for such a period in mole and (ii) the integration period is not the same for the two year and does neither represent a year or a specific season. I suggest instead of presenting cumulative sums to clearly look at the processes and driving factors since there seems to be quite some potential in the dataset to do so.

Due to logistical reasons related to carrying out fieldwork in this remote region of Siberia, the measurements did not cover the whole growing season in both years. The flux interpolation was carried out for 20 and 66 days for each year because all necessary parameters were measured in parallel with fluxes during those time periods and the authors wanted to maximize the interpolation periods. To make the fluxes between two years comparable, the authors now added the average daily flux rates (g C m$^{-2}$ d$^{-1}$) for both years. Of course even with this format, differences related to the length and/or parts of observation periods still have to be considered carefully when interpreting differences, but absolute values are not influenced any more by the length of the time series. However, we decided not to extrapolate data

beyond the observation period, because such an attempt would have been subject to very high uncertainties due to the rapidly changing conditions over the summer season.

(4) The discussion needs substantial improvement concerning structure, scientific argumentation as well as concerning the logic. There are few reference but too often restating of results.

The authors plan to restructure the manuscript by combining results and discussion parts into a single section. Moreover, many paragraphs and sentences will be written more concisely, and references will be added to support our lines of argumentation.

Minor /Technical Comments:

P2,L27: I think the final sentence is not clearly representing your results. You have both a Pro and a Con result so far from the 2013 and 2014 data. Why would this lead to the current conclusion? We agree with the reviewer that the previous wording of the final sentences of the abstract was misleading. It is certainly correct that we found a net increase in $CO_2$ uptake in 2013 within the drained area, while the net $CO_2$ uptake was reduced in 2014. The final statement was based on the fact that the 2013 measurements (20 days) were only representing a single, short section of the summer season, while the longer (66 days) observation record in 2014 is more representative for the net changes in $CO_2$ fluxes over the summer season as a consequence of drainage disturbance. We edited the wording here to make this message clearer.

P2,L33: until which soil depth and what is your definition of the Arctic? It was corrected to avoid confusion.

P2,L41: delete "that is currently stored in deeper" It was corrected as the reviewer suggested.

P2,L42: please try to be specific: Arctic ecosystems or ecosystems in the Arctic. The Arctic is not an ecosystem you may refer to it as a biome or geographical zone. It was corrected as the reviewer suggested.

P2,L44: What about radiation and nutrient supply – both crucial factors in the Arctic too – especially since you mention PAR in your figures Radiation is a part of climate conditions. To make it clearer, the sentence was modified.

P3,L49: why complex? Indeed this may lead to changes but what makes it so complex, try to be more specific We changed the wording to avoid confusion.

P3,L52: One could add an objective here again, otherwise this is all nice information but the reader does not yet know why this background is important The requested information follows in the next paragraphs.

P5,L91: Which conclusion do you draw from your introduction? My suggestion would be to state the need for more studies needed, paired design studies, extrapolation to models according to x,y,z Such information was already given in the following paragraph but a phrase was added to better connect the two paragraphs.

P5,L92ff: this is important but seems a bit lost here - can you incorporate this information at the beginning of the previous paragraph? The previous paragraph described what was missing in the previous studies and the current paragraph described what we could add through this study. Thus, the authors prefer to leave the paragraph structure as it is.

P5,L97: depends on how you define short-term – the historic data refers to four growing seasons of measurements including one experimental season. How about "an initial hydrological manipulation a decade ago" since you have two season only and can hardly use this as a long-term study. It was corrected as the reviewer suggested.

P5,L98: "This study investigates…" It was corrected as the reviewer suggested.

P5,L104: There hasn't been much on frozen season information been provided in the introduction but one of your foci is particularly on frozen season fluxes. Therefore I suggest to expand towards this topic a bit. Some sentences for the frozen-season $CO_2$ fluxes were added as the reviewer suggested.

P6,L112: you state "annual mean" – for which year or which time period is this provided It refers to average conditions within the period 1980 – 2014, and this information was added.

P6,L115: According to the previous papers on this site, the spring flood occurs occasionally. Was this the case for both years of observations? It occurred in both years and this information was added.

P6,L115: You refer to water table depth even though the water table is above the soil – it's a bit contradicting since this does not refer to a depth but rather to a height. It was corrected as the reviewer suggested.

P6,L116: To which period are you exactly referring to with "early growing season"? Can you specify this – see also the comment to Figure 5. It was corrected as the reviewer suggested.

P6,L117ff: please rephrase: Periodic fluctuations approx.. every 5 years but no persistent long-term in air temperature and precipitation could be identified when investigating meteorological data since 1980. It was corrected as the reviewer suggested.

P6,L120: remove the "of" before 2004. It was corrected as the reviewer suggested.

P6,L120: can you please clarify what you mean with " water from the surrounding are". Surrounding what? And what were the effects was only the inner area of the drainage ditch as seen in Figure 2 drained or also the outer area? Some sentences were added to clarify this. Drainage affects both areas inside and outside of the drainage ditch, but certainly the most pronounced effect was observed within the area encircled by the ditch.

P6,L123: how do such short term fluctuation occur? During the daytime WTD decreased by a few cm because of evapotranspiration. However, WTD increased again due to precipitation events and water supply from thawing permafrost. This information is added in the text.

P6,L125: this is an objective and a statement that has been given before – please avoid redundancies and move objective from the methodology section to the introduction  It was corrected as the reviewer suggested.

P6,L127: is that the previously drained site or an new drained site? It is the area which has been drained since 2004. This was rephrased to avoid confusion.

P6,L129: How did you estimate that this area was not affected? There is no direct evidence that the undrained transect was not affected at all. However, based on the vegetation class analyis (WorldView, 2 x 2 m$^2$ resolution) areas around the drainage ditch are dominated by Carex and shrubs–that represent dry areas, and the range of this area does not go beyond 200 m outside the ditch. Thus, ground-based vegetation community structure analysis before and after the drainage as well as larger spatial scale analysis indirectly infer that the drainage effects reached maximum 200 m outside the ditch. Some sentences were added to clarify this.

P6,L129: try to avoid a mix up of the term site and plot/sampling location. I assume you are referring to plots at the two subsites or the one site in Chersky, depending on the fact whether you have seperate experimental fields for instance or other criteria. It was corrected throughout the manuscript.

P7,L133: How do you define representative vegetation? Representative for what? It was corrected to clarify it.

P7,L133: "small enough..." – does that bias your measurements/results? The authors agree that selecting small tussocks may lead to biases concerning the representativeness of our flux data for the larger area although very large trees were not common in this ecosystem. However, we believe that the chosen setup of our chamber system was the optimal solution for the objectives of this field work campaign, and its limitations are acceptable. This potential biases in the flux data were already described in the text.

P7,L134: I suggest to rephrase the sentence towards: "Our analysis comprises the analyis of x weeks of y during .... It was corrected as the reviewer suggested.

P7,L136: you may delete the full sentence. OK.

P7,L138: repharse to: Subsites and plots are labeled according to... It was corrected as the reviewer suggested.

P7,L145: What about soil temperature and soil water measurements in the chamber as well as frozen or not frozen status? Soil temperatures, water table depths as well as thaw depths were measured and described in the section 2.3. (previously section 2.4)

P7, L145: Did you test the chambers to avoid Venturi effects? Our chamber system does not have a vent with a structure that potentially causes Venturi effects (a vent with an additional structure that is parallel to terrain so that fluxes can be affected by wind). Also, chamber fluxes were not affected by wind speed, proving that there was no Venturi effect. However, we did have an opening valve on top of the chamber to avoid pressure effects when placing the chamber onto the collars. This information was added.

P7,L152: which should then equal ER, correct? To my opinion is should be dark during the frozen time most of the time anyway. Some sentences were added to clarify this issue. However, in November when fluxes were measured, it was not dark all the time. Thus, the authors prefer to keep NEE instead of changing it to ER.

P7,L155ff: very good and I can imagine how much work this must have been in this region. ☺

P7,L159: please correct throughout the whole manuscript: _mol m-2 s-1 instead of _mole. It was corrected as the reviewer suggested.

P8,L159: I suggest to add a citation of a typical chamber flux calculation paper A reference was added as the reviewer suggested.

P8,L169: what do you mean with nearby location? It is rephrased to avoid confusion.

P8,L172ff: Why on dry biomass, if you have very "fleshy" plants they could have a larger realistic contribution to the overall biomass then drier plants but if you base this on dryweight only, the effect might disappear. Because water content of plants can vary much by environment, biomass was estimated based on dry mass. This is added in the text.

P9,L179ff: this is unclear. also, how do the two different methods compare? To be able to compare vegetation community structure analyses between 2003 and 2013, we decided to first apply the same method as used by Corradi et al. (described in the text). In the following year, a different method–non-destructive–was applied for each plot so that we could directly link the vegetation community structure within the flux chamber frames with observed $CO_2$ fluxes. To avoid confusion, we changed parts of this paragraph.

P9,L182: Write full words in the heading There were comments from the editor that the abbreviations have to be defined just once when they were mentioned for the first time in the manuscript, while later on only abbreviations should be used. If abbreviations used in the headings are rarely used in the manuscript, they can hinder readers to easily understand. However, abbreviations in the heading were WTD, TD and GPP, which were used very frequently throughout the manuscript, and it is unlikely that readers miss them.

P9,L187: data are always plural. It was corrected throughout the manuscript.

P9,L189ff: phrasing is incorrect – I assume that the co authors that are native speaker could solve this issue. Just as a side note, one could easily reject a paper due to such "technical errors". At the moment it is rather an obvious sign that not all co-the authors have fully read the manuscript. We already clarified our view on the role of co-author contributions for this manuscript in our second statement to the summary text provided by the reviewer. The authors acknowledge that there were some grammatical errors in the manuscript, and we regret that not all of them have been caught by our internal reviewing and proofreading process. However, we do not think it is appropriate to accuse the co-authors of neglecting their responsibilities based on such observations.

P10,L204: why does this approach make it better to compare with 2013 data? The sub-season 3 of 2014 covered a similar period of the field campaign in 2013 after snowmelt. This explanation was added to the text.

P10,L205: can you name these pre, peak, post season and can you do this similar in all papers originating from this research The definition of seasons developed for the research presented herein has been customized for the structure of the chamber flux datasets, i.e. the observation periods covered by this dataset. Accordingly, we plan to use it also for subsequent studies that build on the same dataset.

However, it would not make sense to apply the same scheme for e.g. the eddy-covariance fluxes, which are based on continuously running systems and thus cover much longer time periods.

P10,L207: why august? August was the latest month of the summer season covered within the campaign. Thawing of permafrost soils usually proceeds into early fall, thus within our datasets thaw depths were deepest in that month. The effects of water table depths on thaw depths become more distinct when thaw depths develop, given that differences in water table depths between the two sites begin in early July. This explanation was added in the text.

P10,L210: Please replace "to find out" with "To investigate". It was corrected as the reviewer suggested.

P11,L218ff: This is very confusing, please simplify It was corrected as the reviewer suggested.

P11,L223: what do you mean with -1 . GPP? Because both GPP and ER are positive values, -1 was multiplied to calculate NEE. To avoid confusion, the sentence was edited.

P11,L236: from sensors installed in "the" chamber system. It was corrected as the reviewer suggested.

P12,L239: how did you optimize, via bootstrapping methods? It was optimized by applying a scaling factor between $GPP_{modeled}$ and $GPP_{observed}$ as described.

P12,L247: please remove "equation" All equations were corrected as the reviewer suggested.

P12,L249: measurements. It was corrected as the reviewer suggested.

P12,L251: temperature from the meteorological station. It was corrected as the reviewer suggested.

P12,L254: insert "," between 2104 and while. It was corrected as the reviewer suggested.

P12,L254: numbers below eleven are commonly given in words It was corrected as the reviewer suggested.

P12,L255: please change to Carex sp., Erophorum angustifolium... It was corrected as the reviewer suggested.

P13,L266: wording It was corrected to avoid confusion.

P13,L267ff: name the other terms It was rephrased to clarify.

P13,L269: How did you add both error term? NEE was calculated by subtracting GPP from ER, and accordingly errors of GPP and ER were added in the same way. More detailed explanation was added in the text.

P13,L270ff: how many and why? For bootstrapping, we randomly sampled 80 % of data points. During the 2000 times of sampling, parameters were not acquired for some cases if the data points were chosen only from a certain range of PAR or too scattered to produce light-use efficiency curve. In this case, the authors skipped estimating uncertainty ranges, assuming the number of data points was not enough. However, the authors prefer to leave the sentence as it was because the procedure we applied was described concisely.

P13,L274: time statement goes usually to the end of the sentence. It was corrected as the reviewer suggested.

P13,L275: please explain since these are two antagonistic fluxes. were both fluxes low or large but equal? Assuming that "antagonistic fluxes" refer to the opposing effects of precipitation and evapotranspiration on WTD fluctuations, what we were trying to say here is that we found very little fluctuation in water table levels. There are many factors that increase or decrease water table depths in this ecosystem: precipitation, melting ice from previously frozen soil layers and condensation can increase water table depths while drainage and evapotranspiration can decrease it. However, during this period water table depths were kept generally constant despite minor fluctuations due to aforementioned reasons. We restructured the sentence to point this out more clearly.

P13,L275: please refer to Figure 3 here and for Figure 3 I suggest to state very clearly that one panel is based on 2013 data only and one is 2014 data only! The authors changed the sentence in the text and added a reference to Figure 3. However, the fact that the upper panel was from 2013 and the lower panel was from 2014 was already written in the figure caption, and we see no good option of including data years in the panels themselves in this case.

P13,L279: 2014. However... It was corrected as the reviewer suggested.

P13,L280: but this happened at both sites? Correct. We observed this pattern at both sites but more distinctively in the drained transect. This explanation was added to the text.

P15,L304ff: please simplify. I also wonder why you treat vegetation structure independent from WTD and TD even though these are obviously linked It was simplified as the reviewer suggested. Regarding the comments about vegetation and WTD, the authors tried to explain differences in $CO_2$ fluxes by vegetation community structure and soil temperatures that had been affected by drainage. Thus, it is true that WTD and vegetation community structure are linked, but we differentiated between a reduced WTD as the "primary disturbance" and shifts in vegetation community structure as a "secondary effect". For instance, although undrained_low and drained_low show similar trends in WTD over the growing season, they have different vegetation community structures because of the 10-year drainage history in drained_low. This is why the authors tried to separate effects of vegetation community structure from those of WTD.

P15,L312: That is something that could be indicated in your Figure 1 scheme and make the whole story much clearer This is one of the results that came out through this research, so we would prefer to separate the detailed presentation of this finding from the methods section (where Figure 1 is embedded). Still, to make the readers aware of the implications of drainage disturbance on the soil thermal regime from the start, the authors included additional sentences in the caption of Figure 1.

P15,L318: well done, but why do you give r values and its sufficient to show the numbers either in the figure or the text but there is no need to repeat these. The authors believe that Tables and Figures should be self-explanatory. We therefore included r values in the main body because all results should be mentioned in the results section, and also in the figure to provide an important piece of information to the readers.

P15,L321ff: Fine, but what does this tell us? The authors do not understand what the reviewer is referring to here. If this statement refers to the missing interpretation within the results section in the original

submission, this was based on our choice to separate results from discussion in this version of the manuscript. In the revised version, the result and discussion sections will be combined, and we hope this adequately addresses the criticism of the reviewer.

P16,L326: you might want to consider stating this slightly different-take the maximum uptake as 100% and then just give with > CarexShrub (how many of 100% remained) would there be a difference between CarexEriophorum and EriophorumCarex? Meaning does the order of the names mean that the first was more abundant? your abundance was based on the 10% criterion, so there were no plots with all three plant species? To compare the flux rates among vegetation group in a simpler way, this part will be re-written. The order of the species name does not have any implication here, species names were simply grouped for categorization. Also, there were plots with all three plant types existing, but none of them had a coverage percentage of > 10 % for all three types.

P16,L329: here I suggest a bar diagram, so that the reader is able to follow. The authors decided to change those plots into box plots.

P16,L332: I suggest to not jump between drained and undrained an veg. groups - one after another To make it easier for the readers to follow our line of argumentation, the authors rephrased those sentences.

P16,L339ff: This is redundant Statements given in this section are actually not redundant since the previous paragraph described 2013 data while this paragraph described 2014 data. However, to point out this fact more clearly the authors changed parts of these sentences.

P16: I suggest to find a way, to simplify the drained undrained and species mixtures.... see also the comments given to the figures It was corrected as the reviewer suggested.

P16,L343: soil temperature effects of what? This phrase referred to effects on $CO_2$ fluxes. It was corrected as the reviewer suggested.

P17,L344: why do you explain one Variable with vegetation primarily and the other component with WTD even though both are linked? The purpose of this part was to correlate ER with soil temperatures, not with WTD. Again, we see shifts in WTD as the primary disturbance, which causes changes in soil temperatures as a secondary effect. Therefore, we first needed to separate high- and low-WTD plots, and then analyzed how "changed" soil temperatures affected ER. The authors corrected parts of the text to describe this strategy more precisely.

P17,L349: Table, Figure? It was corrected as the reviewer suggested.

P17,L355ff: I have three general comments here. I think its better to focus on the actual results instead of doing an error analysis, discussion first – this can be done either in the methodology or in the discussion. Secondly you do not have too much data to do a robust gap-filling and base results on the gap-filled data, therefore I suggest to stick with the measurements primarily. Also you provide cumulative flux values for 20 days in 2013 and 66 days in 2014. How relevant are these or how do you suggest one should set this in perspective to other studies? My suggestion would be to remove this part or if this is not possible to focus on specific season and also keep same time intervals in both years. Last but not least cumulative fluxes should be given in g C or CO2 m-2 per time interval not in mol to allow comparison. The authors disagree that flux interpolation should be removed. The number of data points is certainly not as large as

would be provided by e.g. an automatic chamber system or eddy covariance towers, but we are aware of the uncertainty associated with this interpolation, and presented related error estimates in section 3.2.1. Concerning our arguments for the need to analyze two periods of different length within different data years, please refer to our statement within the 'major comments' section above.

P18,L376: PAR was excluded... Not only PAR but all PAR-related terms were excluded.

P18,L373ff: if the model considered vegetation type then of course you will find the same effect as before. Please clarify The picture is not as self-evident as suggested by the reviewer. The model took the vegetation type into account, but vegetation type can still show different responses to environmental controls when fitting fluxes to the predominant conditions within each data year separately. For example, if PAR was generally lower in one growing season than in the other, vegetation types may show different patterns in fluxes.

P18,L378: 0.3mol m-2 20days – that's basically nothing and your overall uncertainty of the approach is larger.

This part of the text was re-written.

P18,L381: how can these be similar- can you calculate the fluxes also for a time period of 20 days? so that they become comparable to 2013? please give g C since mol do not make sense for cumulative fluxes Furthermore I wonder what the key message is if you present 20day or 66day cumulative fluxes. How shall these be compared to other studies, annual or growing season estimates would be much more relevant. All units of flux results were converted into g C-$CO_2$ as the reviewer suggested. Concerning the periods of data analysis, as already stated above we decided against extrapolation beyond the observation period, since biases and/or uncertainties may be very high. Because of this, in our opinion it does not make sense to cover the whole growing season or even one complete year as seen in the Table 1. However, one way to improve the comparability with other studies is to provide also mean flux rates per day, even though also in this case one will have to carefully consider at what part of the season a specific average was derived when comparing to other studies. The authors added this information in the text.

P18,L385: When were these observed? It was November as described in the method section.

P19,L387: replace "by" with "of". It was corrected as the reviewer suggested.

P19,L387: rephrase to "emitted on average four times more CO2 than the undrained site." It was corrected as the reviewer suggested.

P19,L393: Can you the authors state why this is the case? This was previously described in the discussion section, which is now the results and discussion section.

P19,L397: how large was the variability, if its too narrow anyway then why would one test it? Unless the flux rates are constant over time, we cannot subjectively conclude that any statistical analyses are unnecessary. Small variability can be explained by small changes in environmental parameters, and it can be tested with statistical tools.

P19,L398: I think you could provide this amount of information in a brief table or add it do an existing table. Originally the authors included a table with this information but the editor suggested deleting it.

P19,L406ff: the mixing of the two sites and the individual sub-locations which are either wet or dry is still very confusing and I suggest to clearly explain this in the beginning but also make this always clear for the whole manuscript. It was corrected throughout the manuscript.

P20,L409: What is "low WTD of undrained_low…"? It meant low WTD values of undrained_low plots. To avoid confusion, it was rephrased.

P20,L411: Which do you think a location with a low water table in the drained area is quickly affected by precipitation? Isn't that contradictory to your experimental setup where you have a drainage channel to actually drain the water? Undrained_low plots were located at a comparable elevation as drained_low plots within the inner part of drainage ditch. However, the lateral extent of these slightly elevated areas was smaller in the undrained areas, which resulted in horizontal distances to the nearest low-elevated areas was three times smaller compared to the drained areas. With precipitation events, assuming that water laterally flows towards the nearby depressed areas with the same speed in both transects, it takes three times longer to drain water in some drained_low plots due to three times wider areas. The authors clarified these effects in the revised text version.

P20,L412: Do you have a reference proofing that Eriophorum is capable of achieving such water holding? A review reference was added. There are many studies that prove that plants can slow down the water flow physically.

P20,L414ff: this is unclear, please explain and clarify It was rephrased.

P20,L417: please delete this paragraph. It was deleted.

P20,L421: Here you state it's a clear vegetation effect but I suggest to argument differently. You introduce a disturbance such as drainage and this has a follow-up effect e.g. on vegetation and then subsequently the carbon fluxes are affected. Please make sure you have a clear logic in your manuscript. We fully agree with this statement, since this message is what the authors originally intended to present. The parts that were written unclearly were corrected accordingly.

P20,L423: replace "died out" with "extinct" Eriophorum was not extinct but the abundance of it decreased. To avoid misleading, this part was re-written.

P20,L425: That is a very precise result and I wonder how you can determine this so accurately. The following sentence contain the same information, please avoid redundancies. "…with larger shrub abundance in the future." Sentences were edited for clarification. 4 % was presented in the result sections and this finding was based on a comparison between plots covered by only Eriophorum and Eriophorum with 10 % shrubs. But the fact that this difference was not significant was also written in the text.

P21,L435: Appendix 3 – consider that some of your figures will be available in b&w only. The currenty color choice makes it impossible to see the error bars. It was corrected as the reviewer suggested.

P21,L436: "…was a reduced abundance of…" It was corrected as the reviewer suggested.

P21,L438: Please explain why a site that contains Eriophorum will have an even larger decrease in GPP compared to the Carex while you state before that Eriophorum is so productive compared to Carex. This

is contradictory. This part was clarified. It is contradictory when only the presence of *Eriophorum* was considered but the explanation in this section was more focused on the transition stage.

P21,L440ff: Is there any proof for your hypothesis? References are fully missing? Also try to avoid restating results References were added.

P22,L451: prevalence -> occurrence. It was corrected as the reviewer suggested.

P22,L455: Can you estimate how much lower the CO2 flux could be? What is the picture of your results? It is challenging to estimate it because one species can accumulate varying amount of carbon in different environments. However, to make the statement clearer, the sentences were re-arranged.

P22,L459: That is the first time you mentioned Rh specifically, Why were there no specific results on this part presented in the results section? Result and discussion sections will be combined with additional description of this result.

P22,L462: soil surface temperatures instead of surface soil temperatures It was corrected as the reviewer suggested.

P23,L481: Where did you find the Rh increase in your results? $R_h$ in the field was estimated by correcting potential respiration rates measured in soil incubation studies for average soil temperatures at the sites. With combined result and discussion sections, it may be easier for the readers to see results and their explanation.

Figure 1: Even though the authors do a great job in bringing up such scheme about the experiment it seems incomplete. What are the fluxes before drainage and I suggest to modify the size of the arrows to see what you hypothesize. Other effects such as vegetation effects are not visible here. The Figure caption refers to a drainage event. When reading the manuscript this seems not to be an event. I also suggest to use the same terms as used in the text for photosynthesis and ecosystem respiration (GPP, ER) and to include NEE. Both figure and figure caption were corrected for better description.

Figure 2: this is not a schematic but an aerial photograph. How about: Aerial photograph of the site with the schematics of the drained and undrained transects. Names of observation locations are indicated with numbers and the core locations are highlighted in yellow. - I further suggest to explain what core locations are and what was observed at the sampling locations briefly since the figure should be fully self-explanatory without the manuscript text It was corrected as the reviewer suggested.

Figure 3: I suggest to highlight the year for the various panels as well as to indicate the wet and dry locations in the respective transects, since this particular issue may lead to lots of confusion. What are "relative terrain heights"? As previously mentioned, the fact that the upper panel was from 2013 and the lower panel was from 2014 was already written in the figure caption. To avoid confusion, "relative" was removed.

Figure 4: Why not providing common boxplots? Figure caption: Abundance of Betula exilis, Eriophorum angustifolium etc.. It was corrected as the reviewer suggested.

Figure 5: I suggest to the subseasons by name instead of 2014.1 etc.. Why did you choose the exponential interpolation approach? Also name the subseasons in the Figure caption It was corrected as the reviewer

suggested. Concerning the fitting curve, the authors tried several functional forms, and found that exponential fits showed the best agreement with the data, as seen in the Figure.

Figure 6: this is very well explained in the text and I suggest to bring this issue up at an earlier point in the manuscript, since this proofs the concept of your experiment and by these structural changes an influence on the CO2 fluxes becomes relevant. It was corrected as the reviewer suggested.

Figure 7: I suggest boxplots instead of the clouds It was corrected as the reviewer suggested.

Figure 8: Consistently explain the length and a name for the subseason throughout the manuscript It was corrected as the reviewer suggested.

---

## Author Comment (AC2) · 24 Mar 2016

*Comments from the reviewers are in black and responses from the authors are in blue. Responses from the authors are sometimes in the past tense because the authors have already been editing soon after receiving comments from the reviewers.*

**Authors' responses to the Reviewer #2**

The topic of the submission is undoubtedly of interest to the readership of Biogeo-sciences Discussions as it presents some interesting data arising from an experimental manipulation and its impacts upon ecosystem functioning plus gas exchange in a floodplain. Of particular note is the attention to phenomena during the frozen season as well as the thaw season. However, my greatest criticism of the overall project is why investigate CO2 exchange in such temporal and spatial detail and yet completely ignore methane, and for that matter nitrous oxide exchange and changes both spatially and temporally. It is surely the balance of changes between these contrasting greenhouse gases, of differing radiative forcing strength and atmospheric concentrations that is key here? I miss this vital context entirely within the paper as currently submitted.

Please see also our response letter to the review #1 for a statement regarding this topic. The authors agree with the reviewer that methane and nitrous oxide are indeed important greenhouse gases. However, $N_2O$ was not covered within this study, while $CH_4$ results were not included in this manuscript to avoid too much material for a single manuscript. Therefore, the objective of this study is to investigate drainage effects on exclusively the $CO_2$ fluxes, not all important greenhouse gas fluxes or budget/balance. This separation is necessary since we found substantial changes in ecosystem structure that can be linked to significantly dryer conditions in the manipulated areas, which in turn result in complex changes in $CO_2$ cycle processes. For example, we observed significant shifts in vegetation community structure, which substantially altered both GPP and ER, and a detailed analysis of these opposing effects is required to understand the subsequent shifts in NEE. Thus, changes in $CO_2$ fluxes itself is important to research.

The abstract is of a suitable length and is reasonably informative, but I do question the concluding remark. It is not sufficiently explanative in relation to the findings of the experiment.

It was corrected to clarify the intended message.

The remainder of the text is rather lengthy and could be substantially consolidated without losing any impact (in fact making it much more impactful).

Result and discussion sections are to be combined and some paragraphs/sentences will be corrected to make the manuscript more concise.

The number of figures is verging on excessive. They could be perhaps consolidated and some relegated to appendix/supplementary materials, providing the key focus the manuscript currently lacks?

The authors deleted 3 figures in the main text.

Page 2 line 25 onwards (and subsequent incidences) –the use and comparison of cumulative figures for 20 days in year 1 and 66 days in year 2 does not seem to me to be sufficiently clear to the reader.

Due to the limited period of field campaign, the measurements did not cover the whole growing season in both years. The flux interpolation was carried out for 20 and 66 days for each year because all necessary parameters were measured in parallel with fluxes during those time periods and the authors wanted to maximize the interpolation periods. To make the fluxes between two years comparable, the authors added results in the format g C m$^{-2}$ d$^{-1}$ as mean flux rates within the given observation periods. Of course even with this format, differences related to the length and/or parts of observation periods still have to be considered carefully when interpreting differences, but absolute values are not influenced anymore by the length of the time series.

The premise and importance of the study are generally both well explained within the introduction section of the manuscript.

Line 114: 'Reliable Prognosis"? What is this – it requires more careful explanation. Detailed information on this reference is described in reference section.

Line 118: what is the magnitude of such fluctuations? This information was added as suggested by the reviewer.

Line 120: the depth of the drainage ditch was? The total depth of the drainage ditch varied considerably in space, so a precise measurement of mean depth cannot be provided. However, we measured differences in terrain heights between ditch sections and the nearest plots along the transect and differences were minimum 50 cm.

Line 129 'affected' not 'effected'. How do you know that the drainage ditch had no effect at 600 m away – how did you ascertain this? The authors added some sentences to clarify this. Concerning the range of drainage ditch, there is no direct evidence that the undrained transect was not affected at all. However, ground-based vegetation community structure analysis before and after the drainage as well as larger scale analysis (WorldView, 2 x 2 m$^2$ resolution) indirectly infer that the drainage effects reached maximum 200 m outside the ditch. For more information, please see the responses to the reviewer #1.

Line 135: It is not clear to the reader what the rationale was for the 3 weeks sampling in 2013 and then 10 weeks in 2014. This should be made clear. This decision was mostly based on administrative and logistic constraints related to carrying out field work in this very remote part of Siberia. Please see also our comments within the response letter to review #1 regarding this topic.

Line 136: you introduce the term transects but have not done so before m- this is very confusing and should be addressed fully. Likewise, the labelling of transects is poorly defined. It was corrected throughout the manuscript as the reviewer suggested.

Line 141: The PVC collar was installed permanently, but how so and when, and to what depth was the soil isolated? For how long was the collar installed prior to sampling beginning? i.e. how long was there for recovery of the vegetation? Did cutting-in of the collar lead, as in many cases reported elsewhere, and

in my own considerable experience, lead to vegetation damage/death in any case? The collars were installed approximately 15 cm into the ground, 3 weeks before the first flux measurement took place in 2013. During installation, we avoided damaging aboveground vegetation. Of course inserting the collars into the ground included cutting, and accordingly damages to belowground plant parts could not fully be avoided. However, in neither of the following field seasons where these plots were used, we found no noticeable plant damage or death around the collars, indicating that our field installations did not influence the vegetation substantially. Regarding the 2013 measurements, we believe that even if there was minor damage to the belowground plant parts, an equilibration period of three weeks should have provided enough time to avoid major effects on our flux data. Still, the authors agree with the reviewer that such potential implications should be discussed in the manuscript, so this information was added in the text.

Line 154: What evidence do you have that using ice packs effectively worked to keep the temperature constant as you claim? Air temperature was monitored with 1 Hz frequency while measuring fluxes, accordingly we were able to keep track of temperature gradients within the chamber. When temperatures increased more than 1 °C per minute, we started using ice packs by placing them inside the collar to keep temperatures stable. The total number of ice packs was adjusted until we found temperature conditions to remain at a stable level. Only then, the actual flux measurements were started. We added some more explanation in the text to clarify this.

Line 158: the units quoted need attention. The units are converted to g C basis.

Line 165 onwards: Your phrasing is not sufficiently clear here regarding the 'conflict' between choosing core sites based on WTD category This part was corrected to clarify this issue.

Line 170: define and quantify 'nearby' accurately please. It was corrected to make it clearer.

Line 174: It is not clear how the data from 2003 form a reference! The year 2003 represents conditions at our observation site before the drainage ditch was installed. All 2003 plots for vegetation sampling fall within the area that is now drained, and by co-locating sampling spots for vegetation community structure in 2003 and in 2013, we could directly assess the longer-term shifts in vegetation as a consequence of altered hydrologic conditions. Accordingly, 2003 data serve as a reference for pre-disturbance conditions. This issue was clarified in the text.

Lines 175-181: I find the explanation here insufficiently clear. You need to provide the reader with a much clearer explanation of what you did and why! It was clarified.

Lines 188-189: Aerobic incubation of soils – insufficient detail is provided here for this to be repeated. More explanation was added.

Lines 206-207: There is insufficient detail regarding why you chose August as your point of reference. August is the month within our datasets where thaw depths were deepest, and also differences in thaw depth dynamics related to the drainage were most pronounced. Please find a more detailed statement in our response letter to review #1.

Line 223: You do not describe the term -1. GPP. This should be corrected. It was clarified.

Line 269-70: how exactly did you add this error range in each case? Error ranges were calculated in each temperature and PAR bin for ER and GPP, respectively. When fluxes were interpolated over time, error ranges at each point were taken from the corresponding PAR and temperature bin that reflected current condition.

Section 3.3.1. I find this, as currently written, to be overly complicated in structure and terminology. The authors need to simplify their scheme substantially and really pull-out the key take-home points from the data. A key issue for me in this respect is that you are comparing data with different time periods of coverage between years. Surely a direct comparison between same time periods each year would be more helpful to the reader? In addition, your choice of use of cumulative data seems at odds with your choice of units. This requires correction. The authors agree with the reviewer's concerns. This part as well as the whole result and discussion sections will be re-written.

Line 396 onwards: low variability of what exactly? In the discussion section, I find the terminology again confusing – clearing this up substantially earlier-on in the paper and then following this through to the discussion would really help the reader comprehend the key messages much better than at present! Low variability was referring to relatively constant flux rates over time. To make the paragraphs easier to follow, this part was re-written and some sentences were added in the introduction.

Line 428: this is rather a weak initial statement – be more specific. It was corrected as the reviewer suggested.

Line 440: What direct evidence do you have to back-up this clear statement? References were added.

Figures:

(1) A useful conceptual figure, but much better reproduction is required. There seems to be a mis-match between continued drainage in the experiment and 'events' stated in the legend. This must be addressed. Figure legend terms do not match directly those used in the text of the manuscript. It was corrected as the reviewer suggested.

(2) You need to include the measurements taken at each observation site. It was corrected as the reviewer suggested.

(3) OK

(4) Grey points are not sufficiently clear on this figure. SD error bars – n = ? All figures that featured gray dots to represent individual measurements in the original submission were changed to box plot format in the revised manuscript version.

(5) Sub-seasons of 2014 are unclear – see my comments on this issue in the text of the manuscript also. There was no comment on this in the text, so the authors do not have the information that the reviewer is referring to. However, the definition of sub-seasons in 2014, which is used in several figures and tables, has been placed at a more prominent position in the revised manuscript, with references to this description added at all figures where the sub-seasons are still in use.

(6) OK – but why is this Figure 6 – surely this information warrants being at the start of the explanations!!
It was corrected as the reviewer suggested.

(7) Clouds of grey points are hard to distinguish – rethink...such as boxplots for example. These would, I think be much clearer for the reader. It was corrected as the reviewer suggested.

(8) Ditto

(9) OK

(10) OK

(11) Comments as (7) It was corrected as the reviewer suggested.

Appendix figures OK.

There are numerous points in the text where small edits of the correct word are required for clarity of the narrative. There are too many to list here, but a native English speaker should be consulted to address these shortfalls. This is the worst such manuscript in this respect that I have read in some years. Please see also our statement regarding language edits in the response letter to review #1. The revised manuscript version will be checked by a professional language editor to improve overall readability.

---

## Author Response (AR2)

*Comments from the reviewers are in black and responses from the authors are in blue.*

**Authors' responses to the Reviewer #3**

The paper describes an interesting field experiment with interesting and relevant results, but the structure is difficult to follow, which makes the manuscript as a whole a struggle to read. The use of poorly-described acronyms persists, it seems as if the authors have invented their own language for describing the field site. The English usage is fine, but the results and discussion section badly needs restructuring for clarity. It just says things, often without support, and certainly with little internal structure to help the reader. Simply replacing this section with separate results and discussion sections – and (please) removing all extraneous information and assumptions (the list below is partial) would cut many pages from the manuscript and make it a pleasure to read and a valuable contribution to this understudied field.

The authors appreciate valuable comments from the reviewer #3. All the aspects that were pointed out have been edited. For detailed information, please see answers below.

On line 36, low air temperature does not necessarily imply C sink and in the following sentence the carbon accumulation is partly attributable to the permafrost itself making C (largely) unavailable for respiration. It is corrected as the reviewer suggested, by adding "that inhibit the mineralization of soil carbon".

The end of the first paragraph could use references beyond those of Schuur et al. (not that these aren't good references). Some references are added, such as "(Abbott et al., 2016; Koven et al., 2011; Schuur et al., 2008, 2015)"

With respect to the statement on line 50 note http://www.pnas.org/content/112/9/2788.short changed. This reference is added.

One may argue with the statement 'The influence of temperature on CO2 cycling processes in the Arctic has been well studied (Belshe et al., 2013)' given uncertainties with respect to permafrost. It is corrected for clarity.

With respect to the statement on line 99, note http://iopscience.iop.org/article/10.1088/1748-9326/9/8/085004/meta to the extent that the observations noted in this paper can be used as a surrogate for long-term ecosystem changes. This sentence has been removed to avoid confusion.

Awkward wording on line 112: As a continuation of analysis hydrological manipulation. It is corrected as the reviewer suggested, by removing "analysis" from the sentence.

The methods section is written nicely, but what constitutes WTD low? A sentence, "Plots were classified as 'dry' when the average WTD of the growing season was lower than -10 cm." is added and this information was added in Table 2.

Subheader 2.2 should be plural. OK.

If the UGGA was used why are only CO2 flux data presented? The authors focused on the drainage effects on $CO_2$ fluxes in this manuscript. $CH_4$ fluxes were measured in parallel, but the presentation of these results in combination with the $CO_2$ results would add too much material for a single manuscript. Thus, the $CH_4$ results will be presented in a separate paper.

Space between 3 and sigma on line 200. OK.

Personally I find it interesting if not a bit inefficient to parameterize a model usef for remote sensing for surface fluxes. From this standpoint, which references suggest the prescribed temperature parameters? Please use the multiplication sign or nothing at all rather than the dot which may be taken to mean the dot product. Also, how were the MODIS data used? The pixel may be too big to see relevant effects and at any rate to properly use modis the values around any given point should be averaged with it due to uncertainties in reshaping the ellipsoidal return function to a square(ish) pixel.

We took $T_{opt}$ of 20 °C from the literature (Mahadevan *et al.*, 2008), and this value appeared reasonable for the site from plots of $T_a$ vs. GPP.

MODIS MOD13A1 observations of EVI at a 500m, 16-day resolution were downloaded from ftp://ladsweb.nascom.nasa.gov/allData/5/MOD13A1/2014/ (Solano et al., 2010).

Observations were only used for pixels and time-periods which had been flagged as being of the best quality in both MOD13 and in the corresponding MOD09 surface reflectance observations. Both spatial and temporal gap-filling approaches were then used to create a clean and gap-free dataset at a $1 \times 1$ km resolution, which accounts for the issues raised by the reviewer.

From the cleaned dataset, a pixel was selected on the basis of its coordinates: the central latitude and longitude of study sites was located within the selected pixel. It was found that the time-series of EVI values at this pixel agreed well with EVI time-series from neighboring, relatively unmixed and terrestrial pixels. So as to reduce variability in parameters or outputs arising from EVI alone, and to mitigate risks associated with using pixels with high water fractions which would concurrently have dampened seasonal variability in EVI, the same EVI values were applied to generate estimates of GPP at all chamber locations.

'Eriophorum a.' is uncommon usage. Use instead 'E. angustifolium'. These and other errors make me question if all of the coauthors contributed to this version of the manuscript. It is corrected throughout the manuscript.

Please quantify 'EriophorumShrub' and 'CarexShrub'. An explanation may appear possibly around line 405 but it's certainly not quantified in the text. This information is added to Table 2 and in the text.

I'm not sure that bootstrapping is the best way to estimate parameter uncertainty in this case. I recommend noting http://www.fasebj.org/content/1/5/365.full.pdf (http://www.fasebj.org/content/1/5/365.short). The method suggested by the reviewer would be a very good strategy for acquiring accurate uncertainty ranges of each parameter. However, the main purpose of the bootstrapping used within the context of our study was to get uncertainty ranges of interpolated fluxes instead of getting those of each parameter. For this purpose, the authors believe that bootstrapping approach was adequate.

Personally I feel that the results and discussion section would read much more nicely if separated into results then discussion. Section 3.2.2 is a particularly egregious example of a section that is difficult to read in the context of a long combined results and discussion section that should be restructured for clarity. The first version of this manuscript that was submitted for peer-review in Biogeosciences had separated Results and Discussion sections. When revising the manuscript, we decided to merge the two sections in order to avoid repetitions, which were pointed out by one of the reviewers. Although the reviewer #3 mentioned that the merged one can be a better option, we decided to stick to this manuscript structure, but restructure certain sections of 'Results & Discussion' as recommended by the reviewer. Accordingly, the results and discussion are not separated, but they have been substantially modified for better readability.

The paragraph beginning line 355 says little and is not supported by data. This paragraph as well as the whole Section 3.2.2. has been corrected.

The paragraph on line 360 is expository and belongs in the introduction or elsewhere. This has been corrected as well.

The discussion of C:N ratios on line 364 was somewhat surprising given the topic of section 3.2.2: soil temperature and TD effects on CO2 fluxes. This section is substantially changed, focusing on soil temperature and TD effects on $CO_2$ fluxes.

The paragraph beginning on line 403 is confusing in part because of the insistence on using poorly defined acronyms like 'control_low' which probably reflects internal dialogue about these treatments rather than something that a reader can hope to understand. A figure might help, or a table of abbreviations. Abbreviations are added in Table 2, and some ambiguous sentences have been edited for clarity.

The statement on line 426 is qualitative. Statistical analysis results are added: "(GPP: $F = 11.23$, $P < 0.001$, $R_{eco}$: $F = 3.63$, $P < 0.01$)".

What is a 'first vegetation effect' on line 428? It is corrected to "One of the vegetation effects".

What does 'stabilize' mean in the context of line 444. Do plants really ever stabilize? It is changed to "E. angustifolium is fully replaced by Carex sp. and shrubs" and "resistant to disturbances" in the line #424 and #429.

The statement on line 445 doesn't make sense. Note also this notion of 'stabilization on line 449 which continues to not make sense. This paragraph has been substantially changed as well.

Regarding litter added to the soil on line 474, can you be sure? What if shrubs have higher leaf area index? It is corrected to "the proportion of litter added to the soil will decrease accordingly".

The statement on line 504 doesn't make sense. It is corrected to "the replacement of E. angustifolium by Carex sp., more aerobic conditions, and increased soil surface temperature all weakened $CO_2$ uptake and increased $CO_2$ emission".

From the conclusion, the results are nice and simple to follow. It's a shame that the results and discussion section doesn't reflect this. The first couple paragraphs should be the beginning of a restructured discussion section and the last paragraph of the conclusions should serve as a succinct conclusions section. The results and discussion are substantially changed for clarity.

Table 1 is nice. OK.

What does anything in Tables 4 and 5 mean? The authors believe that Table 4 includes important information of this manuscript. Thus, it is left as it is. Table 5 is moved to Supplementary information.

Red and green should not be used simultaneously if avoidable (and it's certainly avoidable) in figures 4, 6, 9, and S1. The color palette is changed as the reviewer suggested.

Figure 9 isn't particularly revealing, it may make sense to study the relationship between changes in air pressure and flux. This plot is changed with X-axis with changes in air pressure and Y-axis with NEE.

**References**

Solano, R., Didan, K., Jacobson, A., and Huete, A.: MODIS Vegetation Indices (MOD13) C5 User's Guide, Terrestrial Biophysics and Remote Sensing Lab, The University of Arizona, available at: http://www.ctahr.hawaii.edu/grem/modis-ug.pdf, 2010.